# 'Intelligent' lockdown, intelligent effects? Results from a survey on gender (in)equality in paid work, the division of childcare and household work, and quality of life among parents in the Netherlands during the Covid-19 lockdown

**Mara A. Yerkes**[1]*, **Stéfanie C. H. André**[2], **Janna W. Besamusca**[4], **Peter M. Kruyen**[2], **Chantal L. H. S. Remery**[5], **Roos van der Zwan**[6], **Debby G. J. Beckers**[3], **Sabine A. E. Geurts**[3]

**1** Department of Interdisciplinary Social Science, Utrecht University, Utrecht, The Netherlands, **2** Institute for Management Research, Radboud University, Nijmegen, The Netherlands, **3** Behavioural Science Institute, Radboud University, Nijmegen, The Netherlands, **4** Department of Sociology, University of Amsterdam, Amsterdam, The Netherlands, **5** Utrecht University School of Economics, Utrecht University, Utrecht, The Netherlands, **6** Amsterdam Institute for Labour Studies, University of Amsterdam, Amsterdam, The Netherlands

☯ These authors contributed equally to this work.

* M.A.Yerkes@uu.nl

## Abstract

### Objective

The COVID-19 pandemic is more than a public health crisis. Lockdown measures have substantial societal effects, including a significant impact on parents with (young) children. Given the existence of persistent gender inequality prior to the pandemic, particularly among parents, it is crucial to study the societal impact of COVID-19 from a gender perspective. The objective of this paper is to use representative survey data gathered among Dutch parents in April 2020 to explore differences between mothers and fathers in three areas: paid work, the division of childcare and household tasks, and three dimensions of quality of life (leisure, work-life balance, relationship dynamics). Additionally, we explore whether changes take place in these dimensions by comparing the situation prior to the lockdown with the situation during the lockdown.

### Method

We use descriptive methods (crosstabulations) supported by multivariate modelling (linear regression modelling for continuous outcomes; linear probability modelling (LPM) for binary outcomes (0/1 outcomes); and multinomial logits for multinomial outcomes) in a cross-sectional survey design.

**Data Availability Statement:** The data underlying the results presented in the study are available from the LISS archive (www.dataarchive.lissdata. nl). DOI: 10.17026/dans-x3d-e4fb

**Funding:** Research material for the Covid19 Gender (In)equality Survey Netherlands (COGIS-NL) study was supported by an ODISSEI (Open Data Infrastructure for Social Science and Economic Innovations) grant (ODISSEI, No award number, Dr. Mara A. Yerkes) to collect data during the COVID-19 pandemic. All authors listed on the paper were recipients of the grant (MY, SA, DB, JB, PK, CR, RvdZ, SG). The grant allowed for the collection of data within the existing LISS panel (the Dutch Longitudinal Internet Studies for the Social Sciences). The LISS panel data (including the COGIS-NL data) are collected by CentERdata (Tilburg University, The Netherlands) through its MESS project funded by the Netherlands Organization for Scientific Research. The authors do not receive funding from either CentERdata or the Netherlands Organization for Scientific Research for the purposes of this study. The funder had no role in study design, data collection and analysis, decision to publish, or preparation of the manuscript. CentERdata provided advice regarding study design for the purposes of survey programming; final decisions on study design were the responsibility of the authors. CentERdata had no role in the data analysis, decision to publish, or preparation of the manuscript.

**Competing interests:** The authors have declared that no competing interests exist.

## Results

Results show that the way in which parents were impacted by the COVID-19 pandemic reflects a complex gendered reality. Mothers work in essential occupations more often than fathers, report more adjustments of the times at which they work, and experience both more and less work pressure in comparison to before the lockdown. Moreover, mothers continue to do more childcare and household work than fathers, but some fathers report taking on greater shares of childcare and housework during the lockdown in comparison to before. Mothers also report a larger decline in leisure time than fathers. We find no gender differences in the propensity to work from home, in perceived work-life balance, or in relationship dynamics.

## Conclusion

In conclusion, we find that gender inequality in paid work, the division of childcare and household work, and the quality of life are evident during the first lockdown period. Specifically, we find evidence of an increase in gender inequality in relation to paid work and quality of life when comparing the situation prior to and during the lockdown, as well as a decrease in gender inequality in the division of childcare and household work. We conclude that the unique situation created by restrictive lockdown measures magnifies some gender inequalities while lessening others.

## Discussion

The insights we provide offer key comparative evidence based on a representative, probability-based sample for understanding the broader impact of lockdown measures as we move forward in the COVID-19 pandemic. One of the limitations in this study is the cross-sectional design. Further study, in the form of a longitudinal design, will be crucial in investigating the long-term impact of the COVID-19 pandemic on gender inequality.

## Introduction

Gender inequality pertaining to the unequal division of tasks and/or resources between men and women is one of the most persistent social problems of the 21st century [1, 2]. This ongoing inequality particularly affects men and women's lives across the dimensions of paid work, the division of childcare and household work [3–6]), and quality of life (including leisure time [7, 8], work-life balance [3, 4], and for those in a relationship, relationship dynamics [5, 6]).

The COVID-19 pandemic and far-reaching measures taken by governments to reduce the spread of the virus have the potential to substantially impact these patterns of gender inequality, especially within families with children. Preliminary evidence suggests that particularly in the Global North, the pandemic further divides an already gendered labour market. Women are overrepresented in public sector occupations such as health care, education, and childcare [9]. In the health care sector alone, three-fourths (76%) of European workers are women [10]. Women are similarly overrepresented in health care and community/social service sectors in the US [11] and Australia [12]. These commonly underpaid and undervalued occupations became essential during the COVID-19 pandemic; consequently, many mothers continue to work outside the home. Simultaneously, other female-dominated industries, such as retail,

accommodation services, and food and beverage service activities, have been disproportion-ately affected by lockdown measures because of the (temporary) termination of their services. Therefore, women more than men are likely to see the greatest job losses and reductions in working hours during the pandemic and its aftermath [9]. Further attempts to slow the spread of COVID-19 include office closures and a sudden increase in working from home, measures which also have the potential to substantially impact patterns of gender inequality. In the US, early estimates suggest half of the workforce worked or is working from home full time during the pandemic [13]. In Europe, more than one third (37%) of employees were working from home at the height of the first wave of the pandemic, although this percentage varies across countries, ranging from nearly 20% in Romania to nearly 60% in Finland. Of those working from home in April 2020, one third were parents with children under the age of 18 [14]. Fur-thermore, childcare centres and schools were temporarily closed in most countries during the first wave and social distancing measures discouraged others, such as grandparents, from car-ing for (grand)children, limiting alternatives to formal or parental care. Care and home-schooling responsibilities thus shifted fully onto parents, which may further impact paid work, the division of childcare and household work, and the quality of life among parents in gender unequal ways.

However, it is not yet clear how the sudden changes introduced by COVID-19 lockdown measures impact these three areas of gender inequality. On the one hand, COVID-19 lock-down measures potentially magnify existing gender inequalities, for example by reaffirming women's caregiving role [15, 16]. Initial evidence from countries such as Australia, the UK, the US, and Germany confirm this effect [17–19]. In these countries, women spend more time each day on care tasks than men, resulting in an increase in time spent on caregiving tasks dur-ing the pandemic. In Australia, Cooper and Mosseri [18] suggest women experience a triple burden during the pandemic: many women face stressful and risky work on the frontlines given their overrepresentation in essential occupations, they are witnessing the greatest losses in jobs and hours, and their unpaid care work at home is increasing. On the other hand, COVID-19 lockdown measures have the potential to reduce existing gender inequalities. Given women's overrepresentation in essential occupations and the concurrent push for non-essential workers to work from home, as well as the absence of care alternatives, families may be forced to (re)negotiate the division of childcare and/or household work [20]. In the US, for example, Carlson, Pepin and Petts [19] show that while women still do more caregiving tasks than men, men are doing more than before the pandemic, and almost no men report doing less.

For a thorough understanding of the impact of the COVID-19 pandemic on gender inequality in paid work, the division of childcare and household tasks, and quality of life, more evidence, particularly based on representative data, is needed from additional countries, such as Canada and/or other European countries, as country context and culture are crucial for understanding social processes [21]. There is significant cross-national variation in lockdown measures and supportive social policies in place [22], making country context vital for under-standing the short and long-term impact of COVID-19 on gender inequality.

The Netherlands presents an interesting case study for furthering this evidence base for two reasons. First, the Netherlands was criticized internationally for its unique and comparatively lenient first-wave lockdown measures, coined by the government as an 'intelligent' lockdown, a concept explained in more detail in the next paragraph [23]. Second, the Dutch case is inter-esting for investigating gender inequality because in comparison to other European countries, the Netherlands consistently scores high on gender equality indices in the domains of work, health and knowledge [24]. Thus, in European perspective, the division of tasks and resources between men and women is seen to be quite equal. Yet these high scores mask persistent and

underlying gender inequality in multiple domains, particularly in relation to paid work and care. For nearly three decades now, the Netherlands is dominated by its 'one-and-a-half earner model': The majority of fathers works full-time, whereas the majority of mothers works part-time and spends significantly more time caring for children and doing household tasks [25–27]. The so-called 'intelligent' lockdown could potentially be a catalyst to change the persistent structural gender inequalities embedded in this Dutch work and family model [28].

## The Dutch lockdown measures

To restrict the spread of the virus, the Dutch government put restrictive measures in place from 15 March until early June 2020. The combination of measures is labelled an 'intelligent' lockdown by the Dutch government for their emphasis on individual responsibility rather than state enforcement, with the intent of minimizing the economic, social and psychological impact of the pandemic. The Dutch approach is viewed as relatively 'soft' in comparison to other European settings [29]. In reality, some measures were strict while others could be considered lenient. Strict measures included the closing of childcare centres, schools and universities; online classes and homework replaced on-site education to prevent student delays. Emergency care and schooling for children aged 0–12 was available only as a last resort to workers in essential occupations (including care (both youth care and social support), childcare, public transport, the food chain (e.g. supermarkets), the transport industry, waste/garbage collection/processing, media and communication, education, emergency services, necessary government processes, farming, and occupations in the funeral industry). Further strict lockdown measures included the prohibition of public gatherings and events, which effectively closed down firms and entrepreneurs in the catering and events sector. Bars, restaurants, hairdressers, gyms, and saunas were closed. Other measures, however, were less strict and open to interpretation. Shops and businesses able to maintain social distancing requirements (i.e., 1.5 meters distance between non-household members) could choose to continue operations on-site at their own discretion. Public spaces remained accessible, public transport remained operational, and people were not required to have written authorization for outings, travel or use of public transport [29]. Citizens were resolutely asked to refrain from unnecessary travel, to avoid crowded areas, and to work from home whenever possible. However, while families retained the possibility to spend time outside in the company of other household members or one non-household member as long as social distancing was maintained, social life effectively came to a standstill and leisure activities were significantly impacted.

## Objective of the study

With no clear end to the COVID-19 pandemic in sight, social science insights into the impact of the pandemic on gender inequalities in paid work, the division of childcare and household tasks, and quality of life are crucial for understanding the effects of the pandemic on societal development and the need for short and long-term policy responses. The purpose of this cross-sectional study is to provide insights into the immediate impact of the first wave of lockdown measures using representative data. We answer the following research question: To what extent did the COVID-19 'intelligent' lockdown impact gender differences in paid work, the division of childcare and household tasks, and quality of life of Dutch parents? In answering this question, we have three objectives: to investigate gender differences in 1) paid work (work location, working days and times, perceived work pressure); 2) the division of childcare and household tasks; and 3) quality of life (leisure, work-life balance, and relationship dynamics). We investigate these differences between mothers and fathers during the lockdown and provide a comparison to the situation prior to the pandemic. Meeting these objectives will

inform labour market and family policies as well as broader social science debates on gender inequality during pandemics.

## Methods

### Study design

This study applied a cross-sectional survey design in which we were primarily interested in exploring gender differences. The survey questionnaire contained items measuring respondents' paid work, the division of childcare and household tasks, and quality of life in April 2020 (i.e., one month following the start of the first lockdown in the Netherlands). The questionnaire also contained retrospective items on the same topics to measure change between the period prior to the COVID-19 lockdown measures and during the lockdown. The questionnaire was administered in the Dutch language and all questions referred to the full set of lockdown measures taken by the Dutch government as described above.

We developed and extensively tested questions for the survey questionnaire using a multi-step approach. In the first step, we developed an initial set of questions tapping into the situation at work and home before and during the lockdown. These questions were established using our knowledge of the literature and existing questionnaires. Subsequently, we focused on removing and/or adding questions based on project discussions, time available within the panel (7 minutes), and by comparing our question base with existing data modules that can be linked to the data we collected. In a final step, multiple rounds of pilot testing led to the refinement and finalization of the questions. The fielded questionnaire contained 26 questions. The final codebook including all questions and response categories is available from the Longitudinal Internet studies for the Social Sciences (LISS) panel archive [30].

### Data collection

The survey questionnaire was administered by CentERdata, located at Tilburg University, the Netherlands, using their LISS panel. The LISS panel is a representative, online survey panel based on a true probability sample drawn by the Dutch National Statistics Office (CBS) from Dutch population registers. There is no self-selection into the sample and households without internet access are provided with the necessary broadband connection and computer if necessary. Refreshment samples are drawn periodically to ensure continued representativeness of the panel. The LISS panel consists of approximately 7,000 individuals (4,000 households). Fieldwork with our survey questionnaire took place between 13 and 28 April 2020, one month following the start of the first lockdown.

**Sample.** The target sample included all LISS panel members in a household with at least one member in paid employment and at least one child under the age of 18 living at home. Based on these inclusion criteria, 1,234 LISS panel members received the questionnaire. With a response rate of 71.3%, the final sample consisted of 868 respondents in 643 households. We excluded a total of 16 respondents from the analyses. 14 respondents were excluded because they did not meet the inclusion criteria; they reported that neither they nor their partner were in paid employment prior to the first COVID-19 lockdown. We excluded a further two respondents because they indicated they were not presently working but had completed the survey questions as if they were working (i.e., both respondents provided invalid data). The final analytic sample (all respondents included in the analysis) consisted of 852 respondents, 748 of whom were in paid work at the time of the survey and all of whom lived with at least one child under the age of 18. Most respondents were in a partnered household; a total of 71 single parents were included in the sample.

## Ethical considerations

This study was evaluated and received ethical approval from the Faculty of Social and Behavioural Sciences from Utrecht University. Ethical approval for data collection rests with CentERdata, the LISS-panel administrator, who requires all respondents to sign a written, online informed consent form before participating in the panel.

## Measurements

We explored three themes before and during the lockdown for both the respondent and their partner. The three themes were paid work, the division of childcare and household tasks, and quality of life. All items were self-assessed and reported by the respondent. Differences between the time period before and during the first COVID-19 lockdown were measured in two ways: First, by the introduction of retrospective items. Respondents were instructed in prompts for each question whether to report on their situation 'prior to the COVID-19 pandemic' or 'right now'. Second, some questions asked respondents to compare their current situation to the pre-COVID-19 situation, querying whether something, e.g. work in the evening hours, happened 'more or less often' than before the COVID-19 lockdown measures.

**Paid work.** First, for both mothers and fathers, we assessed the impact of the first COVID-19 lockdown on paid work, focusing on changes in work location, the timing of work, and perceived work pressure. 'Work location' measured whether respondents and/or their partner worked from home (differentiating between usually working from home and working from home due to COVID-19), worked at the usual location outside the home (differentiating between required and voluntary working outside the home), combined work from home with working at the usual location outside the home, or being temporarily furloughed (at home because there is currently no work to do). Due to small numbers of respondents who always worked from home and those who were furloughed, this variable was recoded into four categories for the multivariate analysis: working (almost) all hours from home (= reference category), working partially from home, at normal workplace by choice, and at normal workplace due to nature of the work.

To measure changes in the timing of work, respondents were asked how often they currently worked on normal workdays, normal days off, evenings, and weekends in comparison to the time period before COVID-19. Respondents answered each of the four items in this matrix on a five-point scale ranging from 'a lot less' to 'a lot more'. To aid the interpretation of the multinomial logistic regressions, each of these variables was recoded from five to three categories: no change in the amount of hours worked during the respective time period (= reference category), fewer hours worked (combined categories of 'a lot less' and 'a little less') and more hours worked (combined categories of 'a little more' and 'a lot more').

Lastly, perceived work pressure measured whether respondents experienced more, the same or less work pressure during the lockdown compared to prior to the lockdown. This was measured using seven non-ordinal response categories. To ease interpretation of the analysis and avoid categories with small numbers of respondents, we recoded this variable into three categories. The items used in the analysis distinguished between the same amount of work pressure as prior to the lockdown (= reference category), less work pressure, and more work pressure. 'Same work pressure' is a combined category of 'not experiencing any work pressure during the lockdown nor prior to the lockdown' and 'the same amount of work pressure now as before'. Less work pressure is a combined category of 'didn't experience any work pressure during the lockdown, but did experience work pressure prior to the lockdown', with 'much less work pressure' and 'slightly less work pressure' compared to prior to the COVID-19 lockdown. Similarly, more work pressure is a combined category of 'slightly more work pressure' and 'much more work pressure' compared to prior to the lockdown.

**The division of childcare and household tasks.** Second, to examine the impact on dynamics at home, we asked respondents about the division of childcare and household tasks before and during the lockdown. Respondents indicated, relative to their partner, how much housework and, in separate questions, how much caregiving tasks (including home schooling and help with homework) they did prior to and during the lockdown. These questions were each measured separately using a 7-point scale ranging from 'I do nearly everything' (1) to 'My partner does nearly everything' (7). Based on a comparison of the situation before and during the lockdown, two new variables were computed, one indicating whether the relative share of the respondent had increased (1 = yes, 0 = no (reference category)) and one indicating whether the relative share had decreased (1 = yes, 0 = no (reference category)). This was done for household tasks and childcare separately.

**Quality of life.** Third, we examined the impact of the lockdown on mothers' and fathers' quality of life, by exploring changes in leisure time, perceived work-life balance, and relationship dynamics (disagreements with partners relating to the location of work, the division of childcare and household tasks, and leisure time). Change in leisure time was measured on a 5-point scale, ranging from having much less leisure time compared to before the lockdown (1), to having much more leisure time than before the lockdown (5). Given the skewed distribution of these data, these five categories were recoded into three categories: (much) less leisure time (a combination of much less and slightly less leisure time), no change in leisure time (= reference category), and (much) more leisure time (a combination of slightly less and much more leisure time).

Work-life balance was measured using an adapted measure from the European Foundation of Living and Working Conditions (EUROFOUND) Quality of Life survey. Two questions asked respondents to report on how easy or difficult it was to combine paid work with care (including home schooling and homework) for the period prior to the COVID-19 lockdown and during the lockdown; responses were measured on a 5-point Likert-scale ranging from (1) very easy to (5) very difficult.

Furthermore, we measured relationship dynamics by asking respondents how often they had disagreements with their partner before the lockdown about five issues: work location, scheduling of working hours, housework, caring for children, and leisure time. The answer categories for these items were (1) never, (2) monthly, (3) sometimes, (4) weekly and (5) almost daily. We measured changes between the pre-lockdown situation and the lockdown by asking respondents to compare the frequency of disagreements on the same five issues during the lockdown compared to the period prior to the lockdown. All five items were measured on a five-point scale ranging from (1) a lot less often to (5) a lot more often. Given the distribution of the data, we recoded this variable to less conflict with my partner (1 = yes, 0 = no (reference category)) and more conflict with my partner (1 = yes, 0 = no (reference category)).

**Covariates.** Our gender measure was included from existing LISS data modules, which applies a binary variable (women = reference category). Non-binary options are not included in these modules. The other covariates included in the analyses were: essential occupation, age, sector, educational level, number of children, and school status of children. Respondents were provided the government list of essential occupations and asked whether they (non-essential = reference category) and/or their partner (partner in non-essential occupation = reference category) worked in an essential occupation. In addition, we controlled for several socio-demographic characteristics, including age (in years, centred on the grand mean), sector ((semi)-public, private sector = reference category, sector unknown), educational level (low: primary or secondary qualifications = reference category; medium: vocational qualifications; high: tertiary education), number of children, and school status of children under 18 living in the household (measured here as children of primary-school age, children

of secondary-school age, children of both primary-school and secondary-school age, and children not attending school = reference category). This last category included both children under school age (0–4) as well as children who already finished secondary school.

Descriptive data on the sample using these measures can be found in S1A Table. Half of all working mothers (56%) were employed in an essential occupation during the lockdown, compared to 34% of fathers. Mothers constituted 65% of all essential workers. We note that based on average marginal effects, Dutch fathers were less likely (16 percentage points) to work in an essential occupation than mothers (see S1B Table). Most essential workers were employed in the (semi)public sector.

Fifty-seven percent of households and 46% of respondents in the sample reported that at least one parent was employed in an essential occupation. Sixty-one percent of these households reported that one parent was an essential worker while the other was either not in employment or employed in a non-essential occupation. The remaining households contained families where both parents were employed in essential occupations (28%) and single-parent households (10%). Almost all parents in our study (94%) indicated their children were at home either full-time (88%) or part-time (6%) as a consequence of the general closure of schools and childcare centres during the measurement period. The remaining 6% of children were not at home, and thus in emergency childcare/school.

## Data analysis

We relied on two methods for data analysis: descriptive methods (percentages based on cross-tabulations; tables reported in online appendices) and multivariate modelling (tables displayed). We used two types of multivariate modelling: multinomial logistic regression and linear probability modelling (LPM). We used multinomial logistic regression for the analysis of paid work dynamics, which include nominal outcomes (three or more unordered categories in the dependent variable). Multinomial logistic models estimate the effect of independent variables on the relative probability of being in one of the multiple categories of the dependent variable compared to the reference category, which is not possible using LPM [31, 32]. We reported results from the multinomial logistic regressions using the more easily interpretable odds ratios rather than logistic coefficients. We used LPM to analyse the division of childcare and household tasks, and the quality of life. LPM is a linear regression model on a binary dependent variable (0/1), which is generally preferable to logistic regression for binary outcomes as odds ratios cannot be interpreted as effect measures in mediation or interaction models in logistic regression [33, 34]. Odds ratios reflect unobserved heterogeneity which can make it difficult to compare across models. LPM yields unbiased and consistent estimates of a variable's average effect on the chance that the outcome occurs. The b-coefficients in LPM can be interpreted as average marginal effects, holding all other measures constant at their means. Note that average marginal effects report the differences in percentage points.

## Results

### Paid work

**Work location.**   Our results show that almost half (49%) of the working parents in the sample report working all (38%) or part (11%) of their hours from home due to the COVID-19 lockdown measures in April 2020 (see S2A Table). In our sample, a small share (5%) of working parents indicate they worked most of their hours from home prior to the lockdown, and continue to do so during the lockdown. In addition, substantial shares (42%) of Dutch parents continue to perform work tasks outside the home during the lockdown. A small group of respondents (7%) report being allowed to work from home but choosing not to do so. The

remainder of the parents not working from home (35% of working parents) indicate they work on location because their work cannot be performed from home. Finally, 4% of parents indicate having a job but being at home without work due to the COVID-19 lockdown measures.

The descriptive analysis on work location shows no clear gender differences (see S2B Table). The multinomial logistic regressions confirm this finding (see Table 1). Work location is primarily associated with respondents' employment in essential versus non-essential occupations. Compared to the reference category of working all hours from home due to COVID-19, respondents in essential occupations are 3.4 times more likely to work part of their hours from home, 8.9 times more likely to work in their normal workplace by choice, and 4.7 times more likely to work in their usual location due to the nature of their work. These effects are the same for mothers and fathers, as indicated by the non-significant interaction terms. Thus, while essential workers are disproportionately women, no additional gender effect is found. A notable exception is that fathers are three times more likely than mothers to work in their normal workplace by choice, compared to working all hours from home, although this effect is only marginally significant.

Variation in work location reflects differences in educational background. Highly educated parents are substantially more likely to relocate their work to the home: 54% work entirely from home and 13% perform at least some of their work from home. In contrast, 65% of low

**Table 1. Multinomial logistic regression model: Gender differences in work location (ref = working (almost) all hours from home).**

| | Working partially from home | | | At normal workplace by choice | | | At normal workplace due to nature of the work | | |
|---|---|---|---|---|---|---|---|---|---|
| | B | S.E | OR | B | S.E | OR | B | S.E | OR |
| Male | 0.413 | (0.403) | 1.511 | 1.120* | (0.587) | 3.064 | 0.079 | (0.304) | 1.082 |
| Essential occupation | 1.235*** | (0.409) | 3.439 | 2.183*** | (0.602) | 8.870 | 1.545*** | (0.313) | 4.686 |
| Male * essential occupation | 0.288 | (0.546) | 1.333 | -0.968 | (0.748) | 0.380 | -0.035 | (0.445) | 0.965 |
| Partner in essential occupation | -0.291 | (0.294) | 0.747 | 0.006 | (0.343) | 1.006 | -0.092 | (0.234) | 0.912 |
| Age (centred) | 0.011 | (0.025) | 1.011 | 0.025 | (0.030) | 1.025 | 0.014 | (0.020) | 1.014 |
| *Sector* | | | | | | | | | |
| Private sector (= ref) | | | | | | | | | |
| (semi-) public sector | -0.468 | (0.332) | 0.626 | -1.433*** | (0.456) | 0.239 | -0.891*** | (0.280) | 0.410 |
| Sector unknown | -0.472 | (0.348) | 0.624 | -0.442 | (0.393) | 0.643 | -0.321 | (0.266) | 0.725 |
| *Educational level* | | | | | | | | | |
| High | -0.496 | (0.623) | 0.609 | -0.896 | (0.637) | 0.408 | -2.701*** | (0.390) | 0.067 |
| Medium | 0.560 | (0.662) | 1.750 | 0.097 | (0.693) | 1.102 | -0.145 | (0.413) | 0.865 |
| Low (= ref) | | | | | | | | | |
| Number of children | -0.055 | (0.185) | 0.947 | 0.293 | (0.217) | 1.340 | 0.110 | (0.150) | 1.117 |
| *School status of children* | | | | | | | | | |
| No children in school | 0.401 | (0.374) | 1.493 | -0.229 | (0.507) | 0.796 | -0.054 | (0.322) | 0.947 |
| Children in primary school (= ref) | | | | | | | | | |
| Child(ren) in secondary school | 0.142 | (0.412) | 1.153 | -0.030 | (0.473) | 0.970 | 0.183 | (0.319) | 1.200 |
| Child(ren) in primary and secondary school | 0.536 | (0.414) | 1.709 | -0.012 | (0.478) | 0.988 | 0.299 | (0.328) | 1.349 |
| Intercept | -1.510 | (0.774) | | -2.572 | (0.921) | | 0.843 | (0.520) | |
| Cox & Snell $R^2$ | 0.332 | | | | | | | | |
| -2LL (df) | 1290.189 (39) | | | | | | | | |
| N | 675 | | | | | | | | |

\* p<0.10

\*\*p<0.05

\*\*\*p<0.01

educated workers report working in their usual location because of the nature of their job; 59% of workers with a post-secondary vocational degree do so as well (see S2C Table). Specifically, parents with tertiary education are 93.3% less likely than their low educated peers to work in their normal location because their work cannot be performed from home compared to working all hours from home.

**Working time adaptations.** Almost two-fifths (38%) of Dutch parents report working less or much less on their normal workdays during the lockdown compared to the situation prior to COVID-19 (see S3A Table). Substantial shares of parents work more or much more in the evenings (40%) or during weekends (31%). Mothers are more likely to adapt their working times: While 49% of fathers report adapting the days on which they normally work during the lockdown, compared to prior to the COVID-19 pandemic, 61% of mothers adapt theirs: 41% percent of mothers work (much) less on their normal workdays and 20% report working (much) more on these days compared to the situation before the COVID-19 pandemic. For Dutch fathers, these percentages are 35% and 14% respectively. Mothers are more likely than fathers (48% versus 31%) to work (much) more on days they would normally have off.

Multinomial logistic regressions show that employment in an essential versus non-essential occupation moderates these gendered work dynamics. As Table 2 shows, fathers are 56% less likely to work (much) less on their normal workdays and 44% less likely to work (much) more on their normal workdays. Fathers' lower relative probability of working less often on normal working days does not apply to those in essential occupations, as evidenced by the positive interaction term. Mothers in essential occupations are also less likely to reduce working on normal working days. The reduction in work on normal workdays is particularly prevalent among mothers employed in non-essential occupations. Fathers' greater tendency to work at normal times is mirrored in their lower relative probability of increasing or decreasing work on normal days off (except for fathers in essential occupations). Net of gender, highly educated parents also experience more changes to the times at which they work (Table 2). Compared to low educated parents, highly educated parents are 3.2 times more likely to reduce and 3.6 times more likely to increase work on their normal workdays. The multinomial logistic regressions for additional adaptations in working times (i.e., working on normal days off, working during evenings and weekends) can be found in S3B-S3D Table.

**Work pressure.** The results demonstrate that the lockdown led to an increase in work pressure for one-third (36%) of Dutch parents; these parents report experiencing more work pressure in April than prior to the lockdown (see S4 Table). Perceived changes in work pressure differ significantly among mothers and fathers, both in terms of who experiences an increase and who experiences a decrease in work pressure. Mothers are significantly more likely than fathers to experience both more work pressure (39% versus 31%) as well as less work pressure (25% versus 19%) during the lockdown than before.

Amongst our covariates, essential occupation matters for perceived work pressure (see Table 3). Parents in an essential occupation are 2.6 times more likely to report increased work pressure during the lockdown, compared to parents not working in an essential occupation. We also find that educational level matters in relation to work pressure. Higher educated parents are more likely to experience changes in work pressure than lower educated parents, with an increased likelihood of experiencing less work pressure as well as more work pressure during the lockdown compared to before.

## The division of childcare and household work

**Division of childcare.** The results of the analysis on the division of childcare show an increase in care tasks for roughly one-fifth of parents: Compared to the situation before the

**Table 2. Multinomial logistic regression model: Adaptations in working times (ref. = no change to amount of work on normal workdays).**

| | Work on normal workdays | | | | | |
| | Less | | | More | | |
| | B | S.E | OR | B | S.E | OR |
|---|---|---|---|---|---|---|
| Male | -0.849*** | (0.253) | 0.428 | -0.589* | (0.356) | 0.555 |
| Essential occupation | -0.720*** | (0.266) | 0.487 | 0.459 | (0.338) | 1.583 |
| Male * essential occupation | 0.890** | (0.376) | 2.435 | 0.235 | (0.477) | 1.265 |
| Partner in essential occupation | 0.037 | (0.198) | 1.038 | -0.377 | (0.258) | 0.686 |
| Age (centred) | -0.024 | (0.018) | 0.976 | 0.017 | (0.022) | 1.017 |
| *Sector* | | | | | | |
| Private sector (= ref) | | | | | | |
| (semi-) public sector | 0.162 | (0.241) | 1.176 | 0.122 | (0.285) | 1.130 |
| Sector unknown | 0.330 | (0.218) | 1.391 | -0.203 | (0.295) | 0.817 |
| *Educational level* | | | | | | |
| High | 1.157*** | (0.327) | 3.179 | 1.287*** | (0.449) | 3.623 |
| Medium | -0.118 | (0.352) | 0.889 | 0.688 | (0.464) | 1.989 |
| Low (= ref) | | | | | | |
| Number of children | 0.010 | (0.128) | 1.010 | -0.041 | (0.160) | 0.960 |
| *School status of children* | | | | | | |
| No children in school | -0.161 | (0.270) | 0.851 | -0.161 | (0.367) | 0.851 |
| Children in primary school (= ref) | | | | | | |
| Child(ren) in secondary school | -0.192 | (0.276) | 0.825 | 0.164 | (0.335) | 1.178 |
| Child(ren) in primary school and secondary school | -0.639** | (0.287) | 0.528 | -0.557 | (0.371) | 0.573 |
| Intercept | -0.215 | (0.449) | | -1.666 | (0.596) | |
| Cox & Snell R² | 0.153 | | | | | |
| -2LL (df) | 1251.600 (26) | | | | | |
| N | 684 | | | | | |

* p<0.10

**p<0.05

***p<0.01

lockdown, 17% of parents report doing a relatively higher share of care tasks during the lockdown (see S5A Table). This share is significantly higher among fathers (22%) than mothers (12%). We also find that a similar percentage of parents (about one-fifth) report doing less care tasks relative to their partner during the lockdown in comparison to the situation prior to the crisis. Here there are only negligible differences between mothers and fathers. While more fathers than mothers report an increase in the relative share of care tasks performed during the lockdown compared to the situation before, the overall division of childcare between mothers and fathers remains unequal. A majority of mothers (60%) reports doing (much) more childcare tasks than their partner during the lockdown (see S5B Table). In contrast, 10% of fathers report doing (much) more childcare tasks than their partner during the lockdown. One-third of parents (34%) reports that prior to the lockdown, childcare tasks were divided more or less equally; during the lockdown this remained about the same (35%).

Based on linear probability model analysis, the increase in fathers' relative share of childcare work is significant (see Table 4). Compared to mothers, fathers have a greater chance (9 percentage points) of reporting an increase in the relative share of childcare tasks performed during the lockdown than before. Furthermore, changes in the division of childcare tasks are related to occupation. Parents who work in an essential occupation had a significantly smaller

**Table 3. Multinomial logistic regression model: Changes in work pressure (ref. = same amount of work pressure as prior to the lockdown).**

| | Less work pressure | | | More work pressure | | |
|---|---|---|---|---|---|---|
| | **B** | **S.E** | **OR** | **B** | **S.E** | **OR** |
| Male | -0.630** | (0.214) | 0.533 | -0.356* | (0.190) | 0.700 |
| Essential occupation | 0.047 | (0.222) | 1.048 | 0.965*** | (0.196) | 2.624 |
| Partner in essential occupation | 0.028 | (0.222) | 1.028 | -0.042 | (0.196) | 0.959 |
| Age (centred) | -0.001 | (0.020) | 0.999 | -0.012 | (0.017) | 0.988 |
| *Sector* | | | | | | |
| Private sector (= ref) | | | | | | |
| (semi-) public sector | -0.123 | (0.266) | 0.884 | -0.244 | (0.232) | 0.784 |
| Sector unknown | 0.033 | (0.242) | 1.033 | -0.209 | (0.223) | 0.811 |
| *Educational level* | | | | | | |
| High | 0.596* | (0.349) | 1.815 | 0.896*** | (0.323) | 2.450 |
| Medium | 0.293 | (0.367) | 1.340 | 0.456 | (0.337) | 1.578 |
| Low (= ref) | | | | | | |
| Number of children | -0.089 | (0.140) | 0.915 | -0.114 | (0.125) | 0.893 |
| *School status of children* | | | | | | |
| No children in school | -0.196 | (0.313) | 0.822 | 0.147 | (0.270) | 1.158 |
| Children in primary school (= ref) | | | | | | |
| Child(ren) in secondary school | 0.014 | (0.296) | 1.014 | 0.051 | (0.271) | 1.052 |
| Child(ren) in primary and secondary school | -0.158 | (0.324) | 0.854 | 0.302 | (0.278) | 1.352 |
| Intercept | -0.529 | (0.469) | | -0.869** | (0.434) | |
| Cox & Snell $R^2$ | 0.082 | | | | | |
| -2LL (df) | 1415.069 (24) | | | | | |
| N | 712 | | | | | |

* p<0.10

**p<0.05

***p<0.01

chance (13 percentage points) of doing more childcare tasks during the lockdown than before. The analysis of decreases in the relative shares of childcare tasks shows the opposite; parents working in essential occupations have a significantly greater chance (14 percentage points) of reporting a decreased relative share of childcare tasks during the lockdown in comparison to parents working in non-essential occupations. Having a partner in a non-essential occupation does not appear to influence whether respondents report doing more or less care tasks during the lockdown compared to before.

**Division of household tasks.** A small proportion of parents (12% in total) report doing more in the household during the lockdown in comparison to before (see S6A Table). Fathers (17%) report this significantly more often than mothers (7%). A total of 13% of parents report doing less household work during the lockdown in comparison to before the lockdown, with only negligible differences between fathers and mothers. Two-thirds (65%) of all mothers reports doing (much) more household work than their partner during the lockdown (see S6B Table). Among fathers, only 10% indicated they did (much) more than their partner during the lockdown. Similar to the division of care tasks, roughly one third (32%) of parents indicates a more or less equal division of household tasks during the lockdown.

In the linear probability analysis, we find that the increase in fathers' relative share is significant (see Table 5). While overall, mothers continue to do relatively more than fathers, fathers have a significantly greater chance (10 percentage points) than mothers of

**Table 4. Linear probability model: Changes in division of childcare tasks.**

| | Increase in relative share of care tasks compared to before the lockdown | | Decrease in relative share of care tasks compared to before the lockdown | |
|---|---|---|---|---|
| | B | S.E | B | S.E |
| Male | 0.089*** | (0.029) | -0.042 | (0.032) |
| Essential occupation | -0.129*** | (0.031) | 0.138*** | (0.034) |
| Partner in essential occupation | 0.023 | (0.030) | -0.029 | (0.033) |
| Age (centred) | 0.001 | (0.003) | -0.004 | (0.003) |
| *Sector* | | | | |
| Private (= ref) | | | | |
| (semi-)public sector | 0.038 | (0.037) | -0.050 | (0.040) |
| Sector unknown | 0.060* | (0.035) | -0.021 | (0.038) |
| *Educational level* | | | | |
| High | 0.074 | (0.048) | -0.072 | (0.052) |
| Medium | 0.043 | (0.050) | -0.062 | (0.055) |
| Low | | | | |
| Number of children | -0.015 | (0.020) | 0.010 | (0.022) |
| *School status of children* | | | | |
| No children in school | -0.030** | (0.041) | -0.038 | (0.045) |
| Children in primary school (= ref.) | | | | |
| Child(ren) in secondary school | -0.106* | (0.043) | -0.053 | (0.047) |
| Child(ren) in primary and secondary school | -0.067 | (0.044) | -0.065 | (0.048) |
| Intercept | 0.167** | (0.066) | 0.266*** | (0.073) |
| Adjusted $R^2$ | 0.058 | | 0.031 | |
| N | 676 | | | |

\* p<0.10

\*\*p<0.05

\*\*\*p<0.01

reporting an increase in their relative share of household work during the lockdown. Considering our control variables, occupation has a larger effect than gender, but only in relation to parents who report doing less housework during the lockdown than before. Parents who work in an essential occupation have a significantly higher chance (11 percentage points) of reporting doing a smaller relative share of housework during the lockdown. Similarly, respondents with a partner working in an essential occupation have a significantly smaller chance (11 percentage points) of reporting doing a smaller relative share of housework during the lockdown.

## Quality of life

**Leisure time.** The results show that almost half (48%) of parents reports having less leisure time during the lockdown than before (see S7 Table). More than half of the mothers (57%) indicates they have less leisure time than prior to the lockdown, in comparison to 36% of fathers. This effect remains significant in the linear probability model analysis (see Table 6). Table 6 shows that parents with an essential occupation face reduced leisure time more often than parents without an essential occupation. Having children in primary school is also a significant explanatory factor for differences in leisure time: Parents with primary school-aged children are significantly more likely to experience having less leisure time during the lockdown than before, in comparison to parents with children in secondary school.

**Table 5. Linear probability model: Changes in division of household tasks.**

| | Does relatively more household tasks | | Does relatively less household tasks | |
|---|---|---|---|---|
| | B | S.E | B | S.E |
| Male | 0.100*** | (0.026) | -0.035 | (0.028) |
| Essential occupation | -0.026 | (0.028) | 0.111*** | (0.029) |
| Partner in essential occupation | 0.021 | (0.027) | -0.105*** | (0.029) |
| Age (centred) | 0.000 | (0.002) | -0.002 | (0.003) |
| *Sector* | | | | |
| Private (= ref) | | | | |
| (semi)-public sector | 0.010 | (0.033) | 0.041 | (0.035) |
| Sector unknown | 0.017 | (0.031) | 0.007 | (0.033) |
| *Educational level* | | | | |
| High | 0.067 | (0.043) | 0.003 | (0.045) |
| Medium | 0.057 | (0.045) | 0.013 | (0.047) |
| Low (= ref) | | | | |
| Number of children | -0.010 | (0.018) | -0.029 | (0.019) |
| *School status of children* | | | | |
| No children in school | 0.022 | (0.037) | 0.027 | (0.039) |
| Children in primary school (= ref) | | | | |
| Child(ren) in secondary school | -0.026 | (0.038) | 0.000 | (0.040) |
| Child(ren) in primary and secondary school | 0.041 | (0.039) | 0.010 | (0.041) |
| Intercept | 0.031 | (0.060) | 0.181** | (0.063) |
| Adjusted R$^2$ | 0.021 | | 0.048 | |
| N | 676 | | | |

* p<0.10

**p<0.05

***p<0.01

**Perceived work-life balance.** Our analysis shows that the deterioration in work-life balance during the lockdown was equally pronounced among mothers and fathers. Few parents (11%) indicate they find it (somewhat to very) difficult to combine work and care before the lockdown (see S8 Table). During the lockdown, however, nearly one-third (29%) of parents perceives the combination of work and care to be somewhat to very difficult, an increase of 18 percentage points. Looking at changes in work-life balance, one-third (34%) of parents reports having greater difficulty combining work and care during the lockdown than before, and only a small group of parents (9%) perceives the current situation as favourable for their work-life balance (See S9 Table).

Once we control for our covariates, gender is insignificant. Rather than gender differences, educational level and the stage of schooling of children appear to explain variation in perceived work-life balance during the lockdown (see Table 7). Based on the average marginal effects, higher educated parents have a significantly greater chance (16 percentage points) to face increased difficulty combining paid work and care during the lockdown in comparison to parents with low (primary or secondary) education. This effect is even larger for parents with children of primary school age. Having primary school age children significantly increases the chance of experiencing difficulty combining work and care (by 28 percentage points).

**Relationship dynamics.** Our final measure of quality of life relates to relationship dynamics: The extent to which partners experience disagreements about the location of work, the division of care or household tasks, or the amount of leisure time. Most parents indicate that prior to the lockdown, they never had disagreements on most of these topics, including where

**Table 6. Linear probability model: Changes in leisure time (ref = no change in leisure time).**

| | Less leisure time | | More leisure time | |
| --- | --- | --- | --- | --- |
| | **B** | **S.E** | **B** | **S.E** |
| Male | -0.163*** | (0.036) | 0.092*** | (0.031) |
| Essential occupation | 0.100** | (0.037) | -0.011 | (0.032) |
| Partner in essential occupation | 0.038 | (0.038) | -0.004 | (0.033) |
| Age (centred) | -0.006* | (0.003) | 0.002 | (0.003) |
| *Sector* | | | | |
| Private (= ref) | | | | |
| (semi-)public sector | 0.021 | (0.045) | -0.047 | (0.039) |
| Sector unknown | -0.016 | (0.042) | -0.026 | (0.036) |
| *Educational level* | | | | |
| High | 0.241** | (0.058) | -0.013 | (0.053) |
| Medium | 0.144*** | (0.061) | 0.001 | (0.051) |
| Low | | | | |
| Number of children | 0.014 | (0.024) | 0.008 | (0.021) |
| *School status of children* | | | | |
| No children at school | -0.087* | (0.051) | 0.026 | (0.044) |
| Children at primary school (= ref.) | | | | |
| Child(ren) at secondary school | -0.289*** | (0.051) | 0.198*** | (0.045) |
| Child(ren) at primary and secondary school | -0.037 | (0.053) | 0.020 | (0.047) |
| No work for me to do | -0.030 | (0.084) | 0.341*** | (0.072) |
| Intercept | 0.381*** | (0.080) | 0.114 | (0.070) |
| Adjusted $R^2$ | 0.158 | | 0.083 | |
| N | 748 | | | |

* $p<0.10$

**$p<0.05$

***$p<0.01$

they or their partner worked (working at the normal work location (79%); working from home (72%)), or the amount of leisure time (56; see S10 Table). However, prior to the lockdown, about half of parents experienced monthly (or more frequent) disagreements about the division of care tasks (51%), and 60% quarrelled about household tasks. The lockdown did not change this situation for most parents. A majority of parents (between 62% and 71%, depending on the measure) reports no change in the prevalence of disagreements in all areas during the lockdown (the normal work location, working from home, caring for children, household chores, and leisure time; see S11 Table).

These relationship dynamics are the same for mothers and fathers, even after including the covariates in the linear probability model analysis. The only significant increase in disagreements is amongst a proportion of parents in relation to the division of childcare tasks. One-fifth of parents reports an increase in disagreements about the division of childcare tasks, with no differences related to gender (see Table 8). However, parents with children in primary school have a higher chance of reporting an increase in disagreements about the division of childcare tasks (13 percentage points) in comparison to parents with children in secondary school.

## Conclusion

Our results show that the Dutch 'intelligent' lockdown, intended to mitigate the health, social and economic impact of the COVID-19 pandemic, significantly impacts Dutch parents. Many parents

**Table 7. Linear probability model: Change in work-life balance.**

| | Easier to combine work and care | | More difficult to combine work and care | |
|---|---|---|---|---|
| | B | S.E | B | S.E |
| Male | -0.032 | (0.022) | -0.034 | (0.036) |
| Essential occupation | -0.023 | (0.023) | 0.045 | (0.037) |
| Partner in essential occupation | -0.019 | (0.024) | 0.048 | (0.037) |
| Age (centred) | 0.000 | (0.002) | -0.002 | (0.003) |
| *Sector* | | | | |
| Private (= ref) | | | | |
| (semi-)public sector | 0.024 | (0.028) | -0.004 | (0.044) |
| Sector unknown | 0.063** | (0.026) | -0.021 | (0.041) |
| *Educational level* | | | | |
| High | 0.002 | (0.036) | 0.163*** | (0.058) |
| Medium | -0.009 | (0.038) | 0.079 | (0.061) |
| Low | | | | |
| Number of children | -0.016 | (0.015) | -0.001 | (0.024) |
| *School status of children* | | | | |
| No children at school | 0.003 | (0.032) | -0.068 | (0.050) |
| Children at primary school (= ref.) | | | | |
| Child(ren) at secondary school | 0.068** | (0.032) | -0.284*** | (0.051) |
| Child(ren) at primary and secondary school | 0.055* | (0.033) | -0.081 | (0.053) |
| Intercept | 0.105** | (0.050) | 0.324*** | (0.079) |
| Adjusted $R^2$ | 0.010 | | 0.080 | |
| N | 748 | | | |

* $p < 0.10$

** $p < 0.05$

*** $p < 0.01$

are experiencing significant changes in paid work (including where and when they work, and subjective work pressure), the division of childcare and household tasks, and quality of life (the amount of leisure time, as well as the perceived ease or difficulty of combining work and care tasks and negotiating about these tasks with their partner during the lockdown). Our study indicates that the impact of the first COVID-19 lockdown measures on Dutch parents is gendered, whereby some existing gender inequalities increase. Below, we summarize our main results.

## Paid work

Mothers adapt the times at which they work, more so than fathers. Mothers not in essential occupations adapt their working times even more, and are more likely to experience increased work pressure during the lockdown than fathers.

## The division of care and household tasks

At home, the division of household and care tasks remains unequal. More mothers than fathers report doing relatively more housework and childcare tasks than their partner, before the lockdown as well as during the lockdown. We also find limited evidence of a reduction in gender inequality in the division of childcare and household tasks. While mothers continue to do more household and caregiving tasks than fathers, the gap decreased somewhat as fathers report doing (somewhat) more during the lockdown than before.

**Table 8. Linear probability model: Change in disagreements about division of childcare.**

|  | Less disagreement about childcare tasks | | More disagreement about childcare tasks | |
|---|---|---|---|---|
|  | **B** | **S.E** | **B** | **S.E** |
| Male | -0.024 | (0.031) | -0.021 | (0.033) |
| Essential occupation | 0.034 | (0.032) | 0.005 | (0.035) |
| Partner in essential occupation | -0.025 | (0.032) | 0.006 | (0.034) |
| Age (centred) | 0.002 | (0.003) | -0.009*** | (0.003) |
| *Sector* | | | | |
| Private (= ref) | | | | |
| (semi-)public sector | -0.080** | (0.038) | -0.043 | (0.041) |
| Sector unknown | -0.058 | (0.036) | 0.067* | (0.038) |
| *Educational level* | | | | |
| High | -0.169*** | (0.050) | 0.168*** | (0.054) |
| Medium | -0.138*** | (0.053) | 0.076 | (0.057) |
| Low | | | | |
| Number of children | -0.005 | (0.020) | -0.034 | (0.022) |
| *School status of children* | | | | |
| No children at school | 0.011 | (0.043) | -0.104** | (0.046) |
| Children at primary school (= ref.) | | | | |
| Child(ren) at secondary school | 0.007 | (0.045) | -0.128*** | (0.048) |
| Child(ren) at primary and secondary school | -0.024 | (0.045) | 0.003 | (0.049) |
| Intercept | 0.360*** | (0.069) | 0.220*** | (0.074) |
| Adjusted $R^2$ | 0.018 | | 0.081 | |
| N | 748 | | | |

* $p < 0.10$
** $p < 0.05$
*** $p < 0.01$

## Quality of life

We do not find that mothers experience a more pronounced overall decline in the quality of life in comparison to fathers. Mothers as well as fathers equally experience a worsening of work-life balance and an increase in disagreements about the negotiation of the division of childcare tasks. However, leisure time, an important indicator of quality of life, decreased much more for mothers than for fathers during the lockdown.

## Discussion

A key strength of the findings presented here is that they provide evidence of the impact of an unprecedented lockdown based on a representative, probability-based sample among Dutch parents with a high response rate, thereby adding much-needed evidence to an international trend which suggests inequalities along gender and class lines are worsening during the Covid-19 pandemic [9, 14, 17–19]. The gender and educational inequalities (a proxy for class inequalities) exacerbated by or brought about by the lockdown measures are a cause for concern. While mothers' greater adaptability need not be problematic in and of itself, the larger decrease in leisure time among mothers and the increased perception of work pressure among mothers in comparison to fathers, is. An initial study by Eurofound similarly suggests workers are sacrificing leisure to meet work demands during the pandemic [14]. Consequently, people with insufficient leisure time experience a poorer quality of life as well as a greater risk of

reduced health and well-being [7]. Greater perceived work pressure and difficulty combining work and care can also have long-term negative effects. Employees who experience high stress, high work load and time constraints for a long period of time are less productive, are more likely to suffer from burnout complaints and are more likely to switch jobs compared to employees who are feeling well [35–37]. This is especially true when employees have insufficient time to recover from work stress on a day-to-day basis [38]. High levels of perceived work pressure, both before and during the lockdown, among mothers and among workers in occupations considered to be essential for society can therefore be worrisome. Particularly as mothers also experience insufficient leisure time, which can be a sign of insufficient recovery time. These concerns are even greater when considered in relation to growing staff shortages in some of the occupational groups where mothers are overrepresented (e.g., care and education).

Meanwhile, although the lockdown measures result in an increase in gender inequality in paid work and some aspects of quality of life in the Netherlands, our study, in line with results from the US [19], suggests some decrease in gender inequality is also occurring. While mothers still do the lion's share of caregiving and household tasks, the increased involvement of fathers in these tasks during the lockdown offers an opportunity and potential catalyst for further erosion of traditional role patterns in the division of household and care tasks. Prior to the lockdown, Dutch couples routinely reported wanting an equal division of work and care, yet fathers remained more likely to participate in full-time paid employment, whereas mothers were more likely to combine a part-time job with taking on a higher caregiving burden [27].

It should be noted that, overall, women appear to be impacted more by job losses resulting from the COVID-19 pandemic in several countries, including the Netherlands [39], as well as in the US, the UK, and Australia. The latter three countries have historically less generous social policies in place to mitigate the consequences of being out of work [17, 18]. Such variation in policy schemes available to support the temporary furloughing of employees may help explain the different effects of lockdown measures in the long run [17, 22]. With or without government support, women are overrepresented in several sectors hardest hit by the lockdown (retail, accommodation services, and food and beverage service activities) and it is expected that these effects will be felt far into the future, further affecting gender inequality [40].

Alongside the gender differences we explore here, our study also raises questions about the effects of working in an essential occupation as well as working from home, especially with regard to quality of life. We find that parents working in an essential occupation experience the highest levels of work pressure during the lockdown, which can result in health-related risks, such as burnout. These workers are also less likely to work from home. Workers in essential occupations, more often mothers than fathers, and the lower educated, largely continue to perform work in their usual workplace. As a result, these workers face larger risks of exposure to the COVID-19 virus, especially if they work in care occupations, use public transport to get to work, or interact closely with colleagues.

Meanwhile, our study also indicates that working from home is reserved primarily for highly educated parents working in public sector, non-essential occupations (i.e. 'knowledge workers'). These parents face potentially larger risks related to the blurring of work and family time, as evidenced by their greater tendency to deviate from their normal workdays and times. Based on our findings, we can speculate about what this might imply for gender inequality. On the one hand, it could be that more opportunities to work from home (and increased flexibility in working hours) will enable mothers to increase their participation in paid work. On the other hand, unfavourable side effects can be expected when we consider our finding that fathers more often choose to go to the workplace during a lockdown situation even when they

could work from home, whereas mothers more often work from home. In the long run, such differences may result in increased gender inequality, with mothers possibly becoming 'online' workers, with potentially fewer career and networking opportunities, and a greater blurring of work and family life. Although working from home can be advantageous to some workers, employers need to anticipate and monitor for potential unfavourable side effects.

## Limitations

We note a number of limitations of our study, primarily related to data availability. Our findings are based on a single country cross-sectional survey design in which we asked respondents to compare their current situation with their situation before the lockdown using self-rated response measures. Cross-sectional data allow for a snapshot of a unique situation, yet longitudinal data are needed to disentangle causal effects and the long-term impact of the pandemic on the unequal division of tasks and/or resources between men and women. Currently, we are collecting additional multi-wave data to investigate the long-term effects of the COVID-19 pandemic on work and family dynamics. A further limitation of our study stems from the absence of data on the age of the youngest child. Consequently, we are unable to distinguish between children not yet of school age and children who have already completed secondary schooling. The distinction made here based on the phase of schooling addresses this issue only partially. Another limitation is the absence of data on the outsourcing of household or childcare tasks. However, it is reasonable to expect that external help would not have a large confounding impact on the results presented here, as outsourcing of such tasks is generally low in the Netherlands [41].

## Future research

The additional evidence base provided by our study offers several avenues for future research. Given the long-term nature of the pandemic, and the continual increase and decrease of measures to counter COVID-19 across countries, future research must consider the long-term effects of such measures. More intensive forms of working from home have been introduced in many countries as a result of the pandemic. Studies need to consider the potential varying effects of (not) working from home for mothers and fathers, especially with regard to their health and quality of life. In particular, the future world of work is expected to enable greater accessibility of working from home.

In the Netherlands and in other countries, whether temporary lockdown measures lead to sustained increases or decreases in the unequal division of childcare and household tasks evident prior to the pandemic remains an important point for research. Within this line of scholarship, attention should focus on the age of children, particularly in order to distinguish the pressure of care tasks for children younger than school age versus children living at home, but no longer attending school. In addition, data on working hours would be useful, particularly in the Dutch case where part-time work is the norm; mothers working part-time might be the ones adapting their work situation more than full-time working mothers. Research on how parents experience the changes in the division of childcare and household work is also needed. Do fathers who take on greater childcare or household tasks enjoy these new roles? Do mothers like these new divisions, or do feelings of discontent arise? Questions such as these become crucial for understanding the potential long-term sustainability of such changes to gender equality patterns.

In addition, while the closure of schools and childcare centres led to greater attention to the needs of parents and families, additional research should consider the societal impact on *all* workers affected by the lockdown measures. For example, the class effects found here extend

beyond parents. Workers on temporary contracts and the self-employed face significant work and income insecurity [14, 17]. Workers in essential occupations experience significant work pressure. Moreover, the impact of lockdown measures can differ across individuals, for example age cohorts. While social distancing requirements cause social networks to fall away for almost everyone, young people experience much higher levels of loneliness during the lockdown than other age cohorts [14]. Additional research could also go beyond the gender binary still common in many datasets, to allow for greater diversity in our understanding of gender inequalities.

## General conclusion

Current developments necessitate the consideration of new (temporary) lockdown measures to curb the COVID-19 pandemic. Clearly, the societal impact of such measures is great. Our study suggests that in potential future lockdowns, policy and research attention should be paid to gender and class differences in the impact of these measures. Attention is needed to mitigate the adverse 'unintelligent' impact of lockdown measures on the health and wellbeing of parents, especially among mothers. For example, should schools need to be closed during future lockdowns, greater support for parents, and more independent assignments for (younger) children may help to mitigate the pressure parents feel when negotiating care tasks with their partner and combining home schooling with work responsibilities. While scientific and policy attention is logically focused on the health risks associated with the COVID-19 virus, the societal implications of attempts to stop its spread must not be forgotten.

## Supporting information

**S1 Table.** A. Descriptive statistics. B. Linear Probability model: Essential occupation.
(DOCX)

**S2 Table.** A. Distribution of employed respondents by work location. B. Distribution of employed respondents by work location and gender. C. Distribution of employed respondents by work location and educational background.
(DOCX)

**S3 Table.** A. Working time adaptations by gender. B. Multinomial logistic regression of shifts in working time (ref. = no change to amount of work on normal days off). C. Multinomial logistic regression of shifts in working time (ref. = no change to amount of work in the evenings). D. Multinomial logistic regression of shifts in working time (ref. = no change to amount of work on the weekend).
(DOCX)

**S4 Table. Perceived work pressure by gender.**
(DOCX)

**S5 Table.** A. Changes in relative share of care tasks. B. Division of care work by gender.
(DOCX)

**S6 Table.** A. Changes in relative share of household tasks by gender. B. Division of household work by gender.
(DOCX)

**S7 Table. Change in leisure time by gender.**
(DOCX)

**S8 Table. Ease or difficulty of combining work and care before and during the lockdown.**
(DOCX)

**S9 Table. Changes in work-life balance.**
(DOCX)

**S10 Table. Never had disagreements with partner on these issues.**
(DOCX)

**S11 Table. No change in disagreements with partner on these issues.**
(DOCX)

**S1 File.**
(DOCX)

## Acknowledgments

We are grateful to Paul de Beer (University of Amsterdam) for his contribution to the initial study and to Joost Leenen of CentERdata (Tilburg University, The Netherlands) for programming the survey.

## Author Contributions

**Conceptualization:** Mara A. Yerkes, Stéfanie C. H. André, Janna W. Besamusca, Peter M. Kruyen, Chantal L. H. S. Remery, Roos van der Zwan, Sabine A. E. Geurts.

**Data curation:** Mara A. Yerkes, Stéfanie C. H. André, Janna W. Besamusca, Peter M. Kruyen, Chantal L. H. S. Remery, Roos van der Zwan.

**Formal analysis:** Mara A. Yerkes, Stéfanie C. H. André, Janna W. Besamusca, Chantal L. H. S. Remery, Roos van der Zwan.

**Funding acquisition:** Mara A. Yerkes, Stéfanie C. H. André, Janna W. Besamusca, Peter M. Kruyen, Chantal L. H. S. Remery, Roos van der Zwan, Debby G. J. Beckers, Sabine A. E. Geurts.

**Methodology:** Stéfanie C. H. André, Janna W. Besamusca, Peter M. Kruyen, Chantal L. H. S. Remery, Roos van der Zwan.

**Project administration:** Mara A. Yerkes.

**Supervision:** Mara A. Yerkes.

**Validation:** Stéfanie C. H. André, Janna W. Besamusca, Peter M. Kruyen, Chantal L. H. S. Remery, Roos van der Zwan.

**Visualization:** Mara A. Yerkes, Stéfanie C. H. André, Janna W. Besamusca, Chantal L. H. S. Remery, Roos van der Zwan.

**Writing – original draft:** Mara A. Yerkes, Stéfanie C. H. André, Janna W. Besamusca, Chantal L. H. S. Remery, Roos van der Zwan.

**Writing – review & editing:** Mara A. Yerkes, Peter M. Kruyen, Chantal L. H. S. Remery, Debby G. J. Beckers, Sabine A. E. Geurts.

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
