## [Decision Letter · Decision Letter 0]

11 Sep 2020

PONE-D-20-22569

Intelligent lockdown, intelligent effects? The impact of the Dutch COVID-19 ‘intelligent lockdown’ on gendered work and family dynamics among parents

PLOS ONE

Dear Dr. Yerkes,

Thank you for submitting your manuscript to PLOS ONE. After careful consideration, we feel that it has merit but does not fully meet PLOS ONE’s publication criteria as it currently stands. Therefore, we invite you to submit a revised version of the manuscript that addresses the points raised during the review process.

ACADEMIC EDITOR: Both the reviewers find a merit in the research question addressed by this paper. I suggest authors to revise the paper according some major comments given by both the reviewers. Apart from the reviewers comments, I advise the authors to improve the flow and readability of the text. Add a limitation section to the study in ordered to address some of the comments of reviewer 1. 

We look forward to receiving your revised manuscript.

Kind regards,

Srinivas Goli, Ph.D.

Academic Editor

PLOS ONE

Journal Requirements:

3.Thank you for including your ethics statement:  "The work presented here has been approved by the Faculty Ethics Review Committee, Faculty of Social and Behavioural Sciences, Utrecht University. Approval number: 20-269.".   

Please provide additional details regarding participant consent. In the ethics statement in the Methods and online submission information, please ensure that you have specified (1) whether consent was informed and (2) what type you obtained (for instance, written or verbal, and if verbal, how it was documented and witnessed). If your study included minors, state whether you obtained consent from parents or guardians. If the need for consent was waived by the ethics committee, please include this information.

4.We noted in your submission details that a portion of your manuscript may have been presented or published elsewhere.

[The descriptive results in this manuscript have been published as part of a policy brief (in Dutch). ]

5.We note that you have stated that you will provide repository information for your data at acceptance. Should your manuscript be accepted for publication, we will hold it until you provide the relevant accession numbers or DOIs necessary to access your data. If you wish to make changes to your Data Availability statement, please describe these changes in your cover letter and we will update your Data Availability statement to reflect the information you provide.

Additional Editor Comments (if provided):

I suggest authors to revise the paper according some major comments given by both reviewers. Apart from the reviewers comments, I advise the authors to improve the flow and readability of the text. Add a limitation section to the study in ordered to address some of the comments of reviewer 1.

Reviewers' comments:

Reviewer's Responses to Questions

**Comments to the Author**

1. Is the manuscript technically sound, and do the data support the conclusions?

Reviewer #1: Partly

Reviewer #2: Partly

2. Has the statistical analysis been performed appropriately and rigorously? 

Reviewer #1: No

Reviewer #2: I Don't Know

3. Have the authors made all data underlying the findings in their manuscript fully available?

Reviewer #1: Yes

Reviewer #2: Yes

4. Is the manuscript presented in an intelligible fashion and written in standard English?

Reviewer #1: Yes

Reviewer #2: Yes

5. Review Comments to the Author

Reviewer #1: Attached as a separate comment document, it explains why the data needs to be relooked as there is a confounder which has a huge potential effect on the outcome of the regression analysis. so authors needs to undertake that additional analysis and explain

Reviewer #2: Manuscript PONE-D-20-22569

General comments

The manuscript claims to explore the impact of the COVID-19 pandemic on gender inequality in work dynamics, family dynamics and quality of life in the Dutch context. It uses an explorative, longitudinal study design and collects both independent and dependant data from a panel. A questionnaire was developed and tested. The included sample consisted of 852 participants. The presented data and results are the first of four waves of longitudinal data.

The strength of the study is that it adds to our understanding of the impact of the COVID-19 pandemic on societal effects, in this study gender inequality. It is indeed an important topic, in light of recent developments worldwide. Particularly, it gives insight in the impact of the pandemic in the Dutch situation. What I like about it is that it emphasizes a very ‘clinically’ relevant subject which is probably recognisable for many people.

While the study appears to be sound, the terminology, methodology and flow is often somewhat unclear, making it difficult to follow. I advise the authors to improve the flow and readability of the text. The major and minor issues are meant to help to make improvements.

Major issues

Grammar

The authors sometimes report in the past tense and sometimes in the present tense. This creates confusion throughout the text. I suggest the authors make improvements.

Title

To bring more focus, it would help to mention the design at the end of the title.

Abstract

The content of the abstract seems not fully summarize the main text. For example, it is difficult to find out where the results are reported and what the final conclusion of the study is. Also, a main limitation was not mentioned.

Introduction

Gender inequality is the main topic of the study, but the authors did not introduce the subject. Also, terminology is often not or not clearly defined. Furthermore, I miss a rationale for a longitudinal study. The final problem here is that the purpose of the study and the objectives (did you have three objectives related to the themes? Or more?) remain unclear.

Method

It remains unclear what type of design was used. I understand that an exploratory, longitudinal study was mentioned in the introduction, but it would be helpful if the authors would start the method section mentioning the design and giving an overview of the design.

Ethical considerations

It is problematic that there is no paragraph concerning “Ethical considerations”. I would suggest the authors to add this paragraph in the method section, after the paragraph/sample.

Intervention

Furthermore, I understand that the authors did not conduct a study using a specific intervention themselves. Nevertheless, “the lockdown measures” may be seen as an intervention. I think it would be helpful if the authors would give a clearer picture of the lockdown intervention of the Dutch government. It might work to do this in the introduction section. I am confused about the questionnaire and the measures. Did the authors develop one questionnaire with multiple measures? It would be helpful if the authors give a clearer description of the applied questionnaire.

Data-analysis

Also, I found it somewhat confusing to find out how data was analysed. I understand there were three main themes (dependent variables) and also independent variables. But how were these analysed? Could the authors help the reader to improve readability here? For example, is it possible to start sentences like, “To analyse… data… we used …”?

Results

Due to the writing style in the results section, it seems that the authors mixed up results and conclusions. The headings and subheadings seem somewhat suggestive and cause confusion about the reported topics. I guess it will help the reader if the authors stay closer to the themes and topics of data-collection and not use question marks.

Discussion and conclusion

The structure and order of this section is somewhat unclear. Again, the summary of the results and conclusions seem to be mixed up or at least it remains unclear what the results are and what the conclusions. Furthermore, it was unclear where and how each result was compared with results from other studies. Finally, I missed a general conclusion.

Minor issues

I have listed my comments in the order of the issues’ appearance in the manuscript.

Title

The title seems not fully in line with the collected data. Particularly the “gendered work” and “family dynamics” seem different. In the main text I found out it was about “paid work dynamics” and “dynamics at home”. Also, I seem to miss quality of life here.

Abstract

Page 1, Line 2 .“This study examines …” seems incomplete. May it also be that the sentence is redundant? I would expect the relevance of the study first.

Line 8. “The question arises to what extent …” seems inappropriate and confusing, because it is not the research question. Could the authors be more specific what they mean here?

Line 9. The design is missing here.

Line 10. “paid work, division of care and household work and quality of life seem not in line with the description in the main text.

Line 11. The specification of quality of life seems redundant here.

Line 12. it would be helpful to split the sentence and describe the analysis method and results separately. Please start a sentence with “Results showed…”

Line 13: It would be appropriate to report a final conclusion here and then refer to the longitudinal study / or further study…

Introduction

Pg 2. Line 27. Since gender inequality is the main subject of the study, authors are recommended to define and operationalize the concept.

Pg 3. Line 57. To start a new paragraph with a question, i.e. “How do the sudden changes …”, does not invite to read on. I would suggest the authors to rewrite the sentence.

Line 74 “For a thorough understanding of … other countries, …” Compared to what countries? US? Please be specific here.

Pg 5. Line 114. I do not understand the relevance of the sentence “Against this backdrop, …”. Please explain in text or maybe delete the sentence?

Line 119. This is the first time that the authors mention the design (exploratory study). To improve readability, it will help if the authors define and give a rationale for an exploratory study here (see also major issues). Also, I think the authors conducted a longitudinal study? I would recommend to add this here.

Line 120. The text “we aim to provide …” in combination with the text in line 122 “We aim to answer the question confuses me. I suggest to describe and distinguish purpose and objectives here. Also, the text in line 123 “… increase or decrease gender inequality in work and family dynamics” seems incomplete compared to the three main themes mentioned in the method section. Please revise the text, to bring the content in line with the method section.

Line 125-126. Again, this list seems not in line with the method section. Please rewrite.

Line 127. The text between brackets seems redundant here. It would be helpful if the authors define the quality of life earlier in the text.

Materials and methods

The order and content of this section confuses me. Below some suggestions.

Pg 6. Line 132. It would help to start with a heading/Method, followed by a subheading “Design” and give an overview of the design here.

Line 135. It seems there is some duplication here. The sample is also explained below. I suggest that the authors delete the description of the sample here.

Line 142. The authors use the term “sample frame” here. In line 145 the authors describe that 16 respondents were excluded. This may suggest that the researchers used inclusion criteria. This seems a bit confusing. If so, I would recommend the authors to be consistent and use the concepts of the “inclusion and exclusion”.

Line 143. “This resulted in” … meaning “a selection of … were included”? Please be consistent here as well.

Line 145. It would be helpful if the authors would add the numbers and specific reasons why these 16 respondents were excluded.

Pg 7. Line 151. The heading “Measures” confuses me. Did the authors use multiple measures? Also, I do not understand why the authors chose not to describe the questionnaire here. I would recommend the authors to describe the questionnaire if possible. I think it would be more appropriate to add the first sentences of the following paragraph (dependent variables) “We explore three themes ….reported by one respondent” to the “Measures” paragraph.

Line 164.

If possible, I suggest it would help the readers if the authors would add another subheading here, i.e. “paid work dynamics” and add subheadings for the other three themes too.

Line 165. To improve readability, it would help if the authors would start this paragraph and sentence with "To explore the impact on paid work dynamics ..." and bring the other themes/dependent variables in line with this.

Line 171. The word “analyses” seems confusing in this context. Please reformulate.

Pg 8. Line 174. I suggest to add a subheading “dynamics at home” and use “to explore”.

Line 175. “ We used relative measures…”. I find this confusing, possibly because the developed questionnaire (which was mentioned before) was not clearly described. I would suggest to make changes to improve understanding.

Line 178. Also, quality of life was a theme and needs a subheading (I think) to improve readability.

Line 181. I understand the Likert-scale. Nevertheless, it would be helpful to describe “what number is good (1) and what is bad (2)” and do the same for the other Likert-scales.

Line 187. The word “Lastly" seems confusing here, suggesting it was a theme. Please rewrite.

Line 192. “We are primarily interested in exploring gender differences”. This sentence seems particularly relevant. I would suggest the authors it should be part of the design paragraph.

Line 193. I found it interesting that the authors used a binary variable. How about gender neutrality and gender diversity ? This may be relevant to mention somewhere in the text/intro/discussion?

Line 194. I guess the word “self-assessment” is redundant here, as it was mentioned before.

Pg 9. Line 207. “Dutch mothers were overrepresented …”. I do not understand the relevance of this sentence here. Would it be more appropriate to add this to the discussion section?

Pg 10. I do not understand why the authors use the heading “Methods” here as it was partly used before. Did the authors mean analyses? That would be more in line with the text. I would like to ask the authors to make improvements here to strengthen the flow of the text. Also, I would suggest to follow the three aforementioned themes in this section, if possible, and describe how each theme was analysed or to be more specific what type of analysis belongs to what. For example, the authors may use sentences like: “To analyse…., … was used”.

Pg 11. Line 250-252. This text seems more a conclusion than a result. I suggest the authors to check and delete these type of texts throughout the results section.

Line 254. The heading seems not appropriate as it is too long and too little in line with the three main themes. I would recommend the authors to follow the main themes for each subheading, use the word “paid work dynamics” here and make further changes for the other subheadings too in the result section, in line with the main themes and topics.

Line 256. It seems strange to start a sentence with a reference in a results section. It feels as if the results are mixed up with the discussion here. Please rewrite.

Line 260, The line “However, it would be misleading…” seems suggestive. Please rewrite.

Pg 12. Line 285-288. Another example where the text seems more a discussion than a presentation of the results. Possibly it is due to the writing style. Again, I would like to ask the authors to check writing style and present the results close to the results of the analyses.

Pg 15. Line 305. As I reader, I realize that the authors studied data concerning the situation before and during the pandemic. Is that right? Then, I guess it would be helpful if the authors improve writing style.

Line 311. The authors mention the type of analysis here, but this was not consistently done throughout the section. I would recommend the authors more consistency in writing.

Pg 17. Line 334. I think that a subheading “Work pressure” would be more appropriate here.

Pg 18. Line 346. The authors mention “other analyses”, what do the authors mean here?

Pg 25. Line 432. “… having less leisure time’. Compared to what? Also, “This decline in leisure time is significantly gendered”. This seems a conclusion.

Pg 28. Line 476. The authors report on “free time here”. I am confused. What is the difference between leisure time and free time?

Discussion and conclusion

Pg. 30. Line 500. Is this where the summary of the results starts? I am confused (see major issues). I would recommend the authors to be more specific about the main results here. The authors could or should consider to follow the main themes of the study.

Line 520. I have some mixed feelings about the sentence “The changes…. For optimism”. Especially the word “optimism” seems confusing here. This may be, because it is not clear to me whether this is still a result or a conclusion. I suggest that the authors rewrite this sentence.

Line 526. I do not understand the relevance or purpose of the question. Please rewrite this sentence to improve the flow of the text. Maybe this question is relevant as an implication in the discussion section?

Pg 32. I am not sure where in the discussion section the authors compared their results with the results of other studies. I recommend the authors to reconsider this matter and rewrite the discussion somewhat so readers can understand the results in a broader context.

Line 576. Future research and limitations seem to be mingled. Please rewrite.

Line 587. The authors discuss the strengths of the study here. I would suggest to mention these earlier in the discussion section before the limitations.

Pg 35. Line 601. “Studies such as ours demonstrate…”. Since it was an exploratory study, I would suggest the authors to formulate be a little more careful here”.

Line 611. I guess the following part of the text are concluding words. I miss a general conclusion, also in line with the purpose of the study.

Final thoughts

I can imagine that the number of comments is somewhat disappointing. I hope the authors can appreciate this as a way to show that the study is an important contributor to the impact of the COVID-19 pandemic and that the comments are intended to make the manuscript easier to read.

6. PLOS authors have the option to publish the peer review history of their article (what does this mean?). If published, this will include your full peer review and any attached files.

Reviewer #1: No

Reviewer #2: No

---

## [Author Response · Author response to Decision Letter 0]

26 Oct 2020

Revisions to PONE-D-20-22569

We have carefully edited the manuscript to ensure it follows all style requirements. This includes a check of all headings, and removing the short title from the title page. We also now first mention the asterisk followed by the pilcrow symbol. 

A title page has been added.

3.Thank you for including your ethics statement: "The work presented here has been approved by the Faculty Ethics Review Committee, Faculty of Social and Behavioural Sciences, Utrecht University. Approval number: 20-269.". 

Please provide additional details regarding participant consent. In the ethics statement in the Methods and online submission information, please ensure that you have specified (1) whether consent was informed and (2) what type you obtained (for instance, written or verbal, and if verbal, how it was documented and witnessed). If your study included minors, state whether you obtained consent from parents or guardians. If the need for consent was waived by the ethics committee, please include this information.

This information has now been included in the ethics statement and amended in the Methods section of the manuscript.

4.We noted in your submission details that a portion of your manuscript may have been presented or published elsewhere.

[The descriptive results in this manuscript have been published as part of a policy brief (in Dutch). ]

The descriptive results (no multivariate analyses) were not formally published but presented in a short “policy brief” in the Dutch language, on the website of Utrecht University. This policy brief was not peer reviewed. Hence, the work submitted to PlosOne does not constitute a dual publication. We note that following submission to PlosOne, we have published a pre-print of the paper on the Social Sciences archive SocArXiv, in line with the guidelines of the journal.

5.We note that you have stated that you will provide repository information for your data at acceptance. Should your manuscript be accepted for publication, we will hold it until you provide the relevant accession numbers or DOIs necessary to access your data. If you wish to make changes to your Data Availability statement, please describe these changes in your cover letter and we will update your Data Availability statement to reflect the information you provide.

The data are published and accessible to the research community after a moratorium of six months. The data used in the paper will become available in November 2020. Should our manuscript be accepted for publication, the data will be available by this time. We will provide the DOIs on the first possible occasion once they are provided to us. 

Additional Editor Comments (if provided):

I suggest authors to revise the paper according some major comments given by both reviewers. Apart from the reviewers comments, I advise the authors to improve the flow and readability of the text. Add a limitation section to the study in ordered to address some of the comments of reviewer 1.

We are pleased to read that both reviewers see merit in the manuscript and that it adds to our understanding of the impact of the COVID-19 pandemic on society. We appreciate the helpful comments provided by the reviewers on our manuscript and this document outlines the changes we have made to incorporate these comments. This includes the addition of a limitation section. We hope the changes outlined here adequately address the concerns of the editor and the reviewers, contributing to a significant improvement of the manuscript. 

Note that to maintain readability of the manuscript, we have not used track changes to show deletions of text, but only to indicate which parts of the manuscript are new/rewritten. 

Reviewer 1

Reviewer 1 expressed the concern that we have potentially overlooked a major confounding variable measuring “alternative care personal in the family, as this will affect both individual as well as relationship with partner and until multivariate regression looks into this the causality cannot be established.” 

In addition, Reviewer 1 notes in line 173 of the manuscript: “having an alternative care giver in the family is a strong dependent variable, neither in sample of variable description it is highlighted, it is important to clarify that, whether in the sample they were excluded and if not their effect needs to needs to be studied as it is a very crucial confounder and without clarification the robustness of regressions can be questioned”.

In the email, Reviewer 1 states: “Attached as a separate comment document, it explains why the data needs to be relooked as there is a confounder which has a huge potential effect on the outcome of the regression analysis. So authors needs to undertake that additional analysis and explain.”

We thank the reviewer for their careful interpretation of the analyses. In response to the point raised by the reviewer, we note two things. First, we work with cross-sectional data, therefore it is not possible to establish causality. We do not attempt to do this in the current manuscript, but rather focus on the relationships established through multivariate analysis. We therefore apologise for any causal language used and have read through the manuscript and adjusted this were necessary. 

Second, it is not entirely clear to us what is meant with “alternative care personal in the family”. Is the reviewer referring to alternative persons who can also provide caregiving, living within the household? Or to other persons who might also need care within the household? 

In the case of the former interpretation, it is uncommon in the Netherlands for multiple generations to live in one household. Hence, the presence of a personal caregiver other than parents (within one household) is extremely rare in the Netherlands. Less than 1% of all households is multigenerational (including children and grandparents) according to data from the EU-SILC (2008). Further, as we state in the manuscript: “Given social distancing requirements, grandparents were discouraged from helping to provide alternative sources of care during the lockdown”. Also, paid personnel in private households in the Netherlands primarily consists of people hired for elderly care or as cleaning staff rather than for childcare, and such help is almost always hired on a part-time basis. Data from 2014 indicate that only 2.3% of the Dutch population either hired nannies or engaged subsidized personalized childcare services (as opposed to institutionalized (formal) services), the latter of which were closed during the first wave of the pandemic. Of the people who rely on domestic childcare services, 53% did so at most once every two weeks. (Bouwens et al. 2014). 

In the case of the latter interpretation, whereby parents may conceivably have informal care duties alongside childcare, the percentage of individuals who have both children and other caregiving responsibilities is low in the Netherlands. Although one in three Dutch people provides informal care, providing more than 8 hours of care a day is limited to 5% of the population. In addition, having young children is a factor that causes people to provide less informal care (SCP, 2014). 

For either interpretation, it is reasonable to expect that alternative care (either being provided by or providing for others) would not have a large confounding impact on the results we are looking at. In other words, while our questionnaire did not include items on paid domestic services, the effects are likely to be limited. Moreover, the presence of childcare personnel in the household, most likely in the shape of a part-time nanny, would be expected to affect the household as a whole, affecting both parents. It, therefore, would not explain the gender differences found in the study. We now include a brief statement on the point raised by the reviewer in the limitations section of our Discussion.

2. Reviewer 1 suggests the title itself has an outcome judgement, that lock down was intelligent- it this what author wants to convey- than it is fine. 

3. Additionally, Reviewer 1 asks to give an overview- even in the abstract- of why we author call it ‘intelligent’- suggesting a definition of it will be helpful.

We refer to an ‘intelligent’ lockdown because this was the official term used for the lockdown measures taken by the Dutch government. The term is meant to contrast the Dutch lockdown, in which efforts were made to limit the economic and social effects of the measures as much as possible, to alternative forms of lockdown, in which measures like shop closures were expected to have larger economic and/or social consequences.

We address this issue by now briefly clarifying this term in the introduction and at greater length in a new section explaining the Dutch context.

4. Line 39 It is an established fact, however, evidence/citation will be helpful.

We now provide additional references to Queisser, Adema & Clarke (2020) for analyses of the effects of COVID-19 on women’s employment across OECD countries and the European Institute for Gender Equality, EIGE, 2019) as well as the US Bureau of Labor Statistics (2020) for evidence about the overrepresentation of women in these sectors and occupations. 

5. Line 44: at gender perspective another important point that can be reflected is the negative effect of their/women's lack of participation. In other words, they also contribute substantially to national economy/GDP, and lack of participation has a direct effect on that.

Reviewer 1 provides an interesting remark about the likely negative impact of women’s job losses and lower participation rates on the economy. We agree with the Reviewer. However, focusing on this remark is beyond the scope of this paper as our argument relates to gender inequality in society and at home rather than any broader macro-economic effects. 

6. Reviewer 1 suggests that the LISS data needs to explained as there can be bias due to the dataset itself. Moreover, at Line 135 Reviewer 1 notes: The findings are regressions analysis are an outcome of how representative LISS panel is, currently it is difficult to assess that, some description may help.

We have substantially reworked the methods section to provide clearer information about the questionnaire we fielded as well as the panel in which it was embedded. The methods section now includes the design of the study, data collection, ethical considerations, and measurements (operationalizations) of all dependent and independent variables.

In this section, we also provide more details on the LISS panel study. “The LISS panel is a representative, online survey panel based on a true probability sample drawn from Dutch population registers by the Dutch National Statistics Office (CBS). There is no self-selection into the sample and households without an internet connection are provided with the necessary broadband connection. Refreshment samples are drawn periodically to ensure continued representativeness of the panel’s sample.”

7. Reviewer 1 comment c) Difference of pre and post covid is not being established through background as well as result section, some relationship demonstration will be extremely important. 

Differences between the time period pre- and post-COVID-19 were measured in two ways: First, by the introduction of retrospective items in the questionnaire. Respondents were instructed in prompts to each question to report on their work and home situations “prior to the Corona crisis” or “at this moment”. Second, some questions asked respondents to compare their current situation to the pre-COVID-19 situation, for example, asking whether work in the evening hours happened “more or less often” than prior to COVID-19 and the lockdown measures. We have added more details about the measurement of respondents’ pre-COVID-19 status and the measurement of respondents’ status during the COVID-19 lockdown in April through questions in which respondents report on their situation “at this moment”. Post-COVID measurements are, unfortunately, not yet possible. In both the methods and the results sections, we have been careful to make the different measurements more explicit.

Reviewer 2

Major issues

Grammar

The authors sometimes report in the past tense and sometimes in the present tense. This creates confusion throughout the text. I suggest the authors make improvements. 

We carefully edited the text to correct this issue. The manuscript is now written in the present tense, with the exception of the methods section and the explanation of lockdown measures in place in April, which we report in the past tense.

Title

To bring more focus, it would help to mention the design at the end of the title.

We have adjusted the title, which now reflects the survey design of our study. “‘Intelligent’ lockdown, intelligent effects? Results from a survey on gender (in)equality in paid work, the division of childcare and household work, and quality of life among parents in the Netherlands during the Covid-19 lockdown”

Abstract

The content of the abstract seems not fully summarize the main text. For example, it is difficult to find out where the results are reported and what the final conclusion of the study is. Also, a main limitation was not mentioned. 

We have re-written the abstract, using sub-headings to ensure we summarize the main text. We have now included a main limitation in the sub-section Discussion on the cross-sectional design of our study.

Introduction

Gender inequality is the main topic of the study, but the authors did not introduce the subject. Also, terminology is often not or not clearly defined. Furthermore, I miss a rationale for a longitudinal study. The final problem here is that the purpose of the study and the objectives (did you have three objectives related to the themes? Or more?) remain unclear. 

We have rewritten the introduction, clarifying any terminology used for readers less familiar with these terms, including gender inequality (defined in the text as the unequal division or distribution of tasks or resources between men and women) and the meaning of gender inequality indices. We have clarified the objectives of the study more clearly at the end of the introduction. We have removed all mentions of a longitudinal study here because the findings we report on here are cross-sectional. 

Method

It remains unclear what type of design was used. I understand that an exploratory, longitudinal study was mentioned in the introduction, but it would be helpful if the authors would start the method section mentioning the design and giving an overview of the design.

We now start the method section with an extra heading, ‘study design’, in which we explain the cross-sectional nature of our survey study. In the long run, we will have longitudinal data – these are currently being collected but are not yet available. However, to avoid confusion, we have taken out all mention of a longitudinal study. We follow the sub-section on study design with a sub-section on data collection. 

Ethical considerations

It is problematic that there is no paragraph concerning “Ethical considerations”. I would suggest the authors to add this paragraph in the method section, after the paragraph/sample. 

This information has been added in a separate sub-section in the methods section.

Intervention

Furthermore, I understand that the authors did not conduct a study using a specific intervention themselves. Nevertheless, “the lockdown measures” may be seen as an intervention. I think it would be helpful if the authors would give a clearer picture of the lockdown intervention of the Dutch government. It might work to do this in the introduction section. I am confused about the questionnaire and the measures. Did the authors develop one questionnaire with multiple measures? It would be helpful if the authors give a clearer description of the applied questionnaire.

We have made the lockdown measures more explicit, bringing them together in one sub-heading on the Dutch situation in the introduction. These can indeed be seen as some type of intervention. In particular, we focus specifically on the government intervention of March 15th, including the closing of schools, childcare centres, and the requirement to work from home as much as possible. 

To measure the impact of this government lockdown intervention, we administered a single questionnaire in April, containing both current and retrospective items investigating respondents’ experiences and conditions before and during the lockdown in April. We have now clarified this issue in the methods section. 

Data-analysis

Also, I found it somewhat confusing to find out how data was analysed. I understand there were three main themes (dependent variables) and also independent variables. But how were these analysed? Could the authors help the reader to improve readability here? For example, is it possible to start sentences like, “To analyse… data… we used …”? 

We have rewritten the methods section (following the description of dependent and independent variables) to clarify the data analysis methods used: descriptive techniques (cross-tabulations) and two multivariate techniques: linear probability modelling, and multinomial logit modelling. We agree that this improves the readability of the paper. 

Results

Due to the writing style in the results section, it seems that the authors mixed up results and conclusions. The headings and subheadings seem somewhat suggestive and cause confusion about the reported topics. I guess it will help the reader if the authors stay closer to the themes and topics of data-collection and not use question marks.

We have rewritten the results section, moving all interpretations to the discussion to make the results section more factual and less suggestive.

Discussion and conclusion

The structure and order of this section is somewhat unclear. Again, the summary of the results and conclusions seem to be mixed up or at least it remains unclear what the results are and what the conclusions. Furthermore, it was unclear where and how each result was compared with results from other studies. Finally, I missed a general conclusion.

We have significantly rewritten these sections. We now start with a separate Conclusion section, providing conclusions for each of the three main themes. We then provide a Discussion section, in which we separate out the key points of discussion, the limitations to our study, areas for future research, and a final general conclusion.

Minor issues

I have listed my comments in the order of the issues’ appearance in the manuscript.

Title

The title seems not fully in line with the collected data. Particularly the “gendered work” and “family dynamics” seem different. In the main text I found out it was about “paid work dynamics” and “dynamics at home”. Also, I seem to miss quality of life here. 

We have changed the title to Intelligent’ lockdown, intelligent effects? Results from a survey on gender (in)equality in paid work, the division of childcare and household work, and quality of life among parents in the Netherlands during the Covid-19 lockdown” 

Abstract

Page 1, Line 2 .“This study examines …” seems incomplete. May it also be that the sentence is redundant? I would expect the relevance of the study first. 

This sentence has been removed. And the objective of the study is now named first.

Line 8. “The question arises to what extent …” seems inappropriate and confusing, because it is not the research question. Could the authors be more specific what they mean here? 

We have changed this sentence to clarify: Given gender inequality existent prior to the pandemic, particularly among parents, it is crucial to investigate the societal impact from a gender perspective.

Line 9. The design is missing here. 

We have added information on the cross-sectional survey design here.

Line 10. “paid work, division of care and household work and quality of life seem not in line with the description in the main text. 

We have carefully gone through the text to ensure this description matches. The manuscript focuses on paid work (where and when men and women work and the work pressure they experience), how they divide childcare and housework, and their quality of life as measured by differences in leisure time, work-life balance, and relationship dynamics, as noted in the parentheses behind the sentence on Line 10. 

Line 11. The specification of quality of life seems redundant here. 

As quality of life can be interpreted in multiple ways, and has different dimensions, we have chosen to describe the three dimensions used here. See also our response to the point at Line 10.

Line 12. it would be helpful to split the sentence and describe the analysis method and results separately. Please start a sentence with “Results showed…” 

We have added sub-headings to the abstract to clarify, and now begin the sub-heading ‘Results’ with the sentence “The results show…” Note that in accordance with previous comments from the reviewers, we have written the results section in the present tense.

Line 13: It would be appropriate to report a final conclusion here and then refer to the longitudinal study / or further study…

We have now included a sub-heading “Conclusion” where we present the conclusion and one sentence on further research.

Introduction

Pg 2. Line 27. Since gender inequality is the main subject of the study, authors are recommended to define and operationalize the concept. 

This concept has been defined here in the introduction as: “the unequal division or distribution of tasks or resources between men and women”. The operationalization can be found in the methods section and pertains to the three themes 1) paid work 2) division of care and household work and 3) quality of life. 

Pg 3. Line 57. To start a new paragraph with a question, i.e. “How do the sudden changes …”, does not invite to read on. I would suggest the authors to rewrite the sentence. 

This sentence has been rewritten.

Line 74 “For a thorough understanding of … other countries, …” Compared to what countries? US? Please be specific here. 

We have specified that we are referring to countries not yet reported on, such as Canada and/or other European countries.

Pg 5. Line 114. I do not understand the relevance of the sentence “Against this backdrop, …”. Please explain in text or maybe delete the sentence? 

We have rewritten this sentence and restructured this paragraph, which we feel helps to clarify the relevance. “The so-called ‘intelligent’ lockdown could potentially be a catalyst to change the persistent structural gender inequalities embedded in this Dutch work and family model [28].”

Line 119. This is the first time that the authors mention the design (exploratory study). To improve readability, it will help if the authors define and give a rationale for an exploratory study here (see also major issues). Also, I think the authors conducted a longitudinal study? I would recommend to add this here. 

We have removed the reference to the exploratory design here to avoid confusion. The cross-sectional survey design is clarified in the abstract and in the methods section. We report on the cross-sectional survey design here and will collect longitudinal data (more waves) in the future.

Line 120. The text “we aim to provide …” in combination with the text in line 122 “We aim to answer the question confuses me. I suggest to describe and distinguish purpose and objectives here. Also, the text in line 123 “… increase or decrease gender inequality in work and family dynamics” seems incomplete compared to the three main themes mentioned in the method section. Please revise the text, to bring the content in line with the method section. 

We appreciate the reviewer’s careful reading of the text here, allowing for clarification, and carefully distinguishing purpose and objectives. We have removed any mention of the term ‘family dynamics’, referring explicitly to the division of household and childcare work throughout the text. We have also rewritten the research question: To what extent did the COVID-19 ‘intelligent’ lockdown impact gender differences in paid work, the division of childcare and household work, and quality of life of Dutch parents?

Line 125-126. Again, this list seems not in line with the method section. Please rewrite.

We have improved the structure in the whole manuscript, including on these lines.

Line 127. The text between brackets seems redundant here. It would be helpful if the authors define the quality of life earlier in the text. 

This has been removed and we report on the three dimensions of quality of life in the abstract.

Materials and methods

The order and content of this section confuses me. Below some suggestions. 

We thank the reviewer for these helpful suggestions. The methods section has been rewritten to take these suggestions into account.

Pg 6. Line 132. It would help to start with a heading/Method, followed by a subheading “Design” and give an overview of the design here. 

We have added this subheading and an overview of the study design.

Line 135. It seems there is some duplication here. The sample is also explained below. I suggest that the authors delete the description of the sample here. 

We have removed this description here.

Line 142. The authors use the term “sample frame” here. In line 145 the authors describe that 16 respondents were excluded. This may suggest that the researchers used inclusion criteria. This seems a bit confusing. If so, I would recommend the authors to be consistent and use the concepts of the “inclusion and exclusion”.

We have removed the mentioning of the sample frame here and use inclusion criteria. We also explain why the 16 respondents were excluded. We retain the mention of the analytical sample, which refers to all respondents in the analysis. We explain this term for purposes of clarity.

Line 143. “This resulted in” … meaning “a selection of … were included”? Please be consistent here as well.

We have changed this into “This selection consisted of 1,234 panel members”.

Line 145. It would be helpful if the authors would add the numbers and specific reasons why these 16 respondents were excluded.

We now explain why these respondents were excluded. In most cases (14) this was because they did not meet the inclusion criteria, as these respondents had not been in paid employment prior to the Covid-19 outbreak. This can happen when the respondents are marked as matching the sample inclusion conditions in the LISS-panel, but self-report their employment status differently in the questionnaire we applied (usually because their employment status has changed since the most recent questionnaire they completed). 

In two cases, respondents were excluded because of comments flagging their replies as invalid. The survey included an open question at the end, in which respondents were encouraged to leave any comments and remarks about the survey. The two respondents in question noted that they were currently not working due to pregnancy leave or parental leave, but said they had filled in all questions as if they weren’t on leave. Hence, data from these cases were excluded for being invalid.

Pg 7. Line 151. The heading “Measures” confuses me. Did the authors use multiple measures? Also, I do not understand why the authors chose not to describe the questionnaire here. I would recommend the authors to describe the questionnaire if possible. I think it would be more appropriate to add the first sentences of the following paragraph (dependent variables) “We explore three themes ….reported by one respondent” to the “Measures” paragraph.

We administered a survey questionnaire that contained multiple measurements. To clarify, and to avoid any potential confusion with the lockdown measures, we have changed the heading here accordingly. In this Measurements sub-section, we describe the questions we administered in the questionnaire and start with the sentence mentioned by the reviewer. We included additional sub-headings for the three themes and for the covariates. Finally, we include a reference to the codebook, including the questionnaire, in the running text.

Line 164.

If possible, I suggest it would help the readers if the authors would add another subheading here, i.e. “paid work dynamics” and add subheadings for the other three themes too.

We have introduced sub-headings for all three themes.

Line 165. To improve readability, it would help if the authors would start this paragraph and sentence with "To explore the impact on paid work dynamics ..." and bring the other themes/dependent variables in line with this.

We have included this sentence under all three sub-headings.

Line 171. The word “analyses” seems confusing in this context. Please reformulate.

We have changed this into “variables”. We meant that we conduct separate analyses for working more or less hours on normal workdays, normal days off, evenings, and weekends. 

Pg 8. Line 174. I suggest to add a subheading “dynamics at home” and use “to explore”.

We have added these subheadings and used the term “to explore” as suggested by the reviewer.

Line 175. “We used relative measures…”. I find this confusing, possibly because the developed questionnaire (which was mentioned before) was not clearly described. I would suggest to make changes to improve understanding.

We now provide additional information on the questionnaire and how it was developed. In relation to this particular issue, we now explain: “Respondents indicated, relative to their partner, how much housework and, in separate questions, how much caregiving tasks (including home schooling and help with homework) they did prior to and during the lockdown. These questions are each measured separately using a 7-point scale ranging from ‘I do nearly everything’ (1) to ‘My partner does nearly everything’ (7)”. We hope this improves understanding. 

Line 178. Also, quality of life was a theme and needs a subheading (I think) to improve readability.

We agree with the reviewer on this point and have added this subheading.

Line 181. I understand the Likert-scale. Nevertheless, it would be helpful to describe “what number is good (1) and what is bad (2)” and do the same for the other Likert-scales.

We have included this in all relevant places in this section.

Line 187. The word “Lastly" seems confusing here, suggesting it was a theme. Please rewrite.

We have changed this into “Furthermore”.

Line 192. “We are primarily interested in exploring gender differences”. This sentence seems particularly relevant. I would suggest the authors it should be part of the design paragraph.

We have added this to the sub-section on study design. 

Line 193. I found it interesting that the authors used a binary variable. How about gender neutrality and gender diversity? This may be relevant to mention somewhere in the text/intro/discussion?

We fully agree with the reviewer on this point. Unfortunately, the LISS-panel works with pre-existing socio-demographic variables, which rely on a binary coding of gender. There currently is no option to report one’s gender identity (including non-binary identification). We have added a point on this in the discussion, in the sub-section on future research.

Line 194. I guess the word “self-assessment” is redundant here, as it was mentioned before.

We have deleted this word.

Pg 9. Line 207. “Dutch mothers were overrepresented …”. I do not understand the relevance of this sentence here. Would it be more appropriate to add this to the discussion section? 

We have deleted this sentence here.

Pg 10. I do not understand why the authors use the heading “Methods” here as it was partly used before. Did the authors mean analyses? That would be more in line with the text. I would like to ask the authors to make improvements here to strengthen the flow of the text. Also, I would suggest to follow the three aforementioned themes in this section, if possible, and describe how each theme was analysed or to be more specific what type of analysis belongs to what. For example, the authors may use sentences like: “To analyse…., … was used”.

We have changed the heading into ‘Data analysis’ which is indeed more appropriate. We have also changed the structure of this section, explicitly following the three themes (paid work, division of childcare and household work, and quality of life) and described what type of analysis was used for each theme. We feel this improves the flow of the text.

Pg 11. Line 250-252. This text seems more a conclusion than a result. I suggest the authors to check and delete these type of texts throughout the results section.

We have carefully edited the text to ensure any conclusions are removed from the results section and moved to the conclusion.

Line 254. The heading seems not appropriate as it is too long and too little in line with the three main themes. I would recommend the authors to follow the main themes for each subheading, use the word “paid work dynamics” here and make further changes for the other subheadings too in the result section, in line with the main themes and topics. 

We have changed the subheadings to shorten and clarify, in line with the main themes and topics.

Line 256. It seems strange to start a sentence with a reference in a results section. It feels as if the results are mixed up with the discussion here. Please rewrite.

We have rewritten this sentence.

Line 260, The line “However, it would be misleading…” seems suggestive. Please rewrite.

We have deleted this sentence, as it was redundant.

Pg 12. Line 285-288. Another example where the text seems more a discussion than a presentation of the results. Possibly it is due to the writing style. Again, I would like to ask the authors to check writing style and present the results close to the results of the analyses.

We have rewritten this part and moved all interpretation to the discussion. 

Pg 15. Line 305. As I reader, I realize that the authors studied data concerning the situation before and during the pandemic. Is that right? Then, I guess it would be helpful if the authors improve writing style.

We have asked the respondents about their work hours before and during the lockdown. We now explain this more clearly in the text.

Line 311. The authors mention the type of analysis here, but this was not consistently done throughout the section. I would recommend the authors more consistency in writing.

We have carefully edited the text to ensure this is done consistently throughout.

Pg 17. Line 334. I think that a subheading “Work pressure” would be more appropriate here. 

This subheading has been changed in line with the reviewer’s comments.

Pg 18. Line 346. The authors mention “other analyses”, what do the authors mean here?

We were referring to the analyses on the other variables. We have deleted this sentence to avoid confusion.

Pg 25. Line 432. “… having less leisure time’. Compared to what? Also, “This decline in leisure time is significantly gendered”. This seems a conclusion.

This is in comparison to the situation prior to the lockdown. We have added “…than before the lockdown” to clarify. The statement that the decline is significantly gendered is referring to the finding that the gender interaction is significant. We have changed this sentence to avoid it reading like a conclusion. 

Pg 28. Line 476. The authors report on “free time here”. I am confused. What is the difference between leisure time and free time? Leisure time and free time were being used interchangeably. We have deleted any references to free time to avoid confusion.

Discussion and conclusion

Pg. 30. Line 500. Is this where the summary of the results starts? I am confused (see major issues). I would recommend the authors to be more specific about the main results here. The authors could or should consider to follow the main themes of the study.

As suggested by the reviewer, we have separated the conclusion and discussion sections. In the conclusion, we provide a summary of results across the three main themes of the study.

Line 520. I have some mixed feelings about the sentence “The changes…. For optimism”. Especially the word “optimism” seems confusing here. This may be, because it is not clear to me whether this is still a result or a conclusion. I suggest that the authors rewrite this sentence.

We rewrote the conclusion and discussion sections to disentangle our interpretations, limitations, and suggestions for future research. We also rewrote this particular sentence, avoiding the word ‘optimism’.

Line 526. I do not understand the relevance or purpose of the question. Please rewrite this sentence to improve the flow of the text. Maybe this question is relevant as an implication in the discussion section?

Longitudinal research needs to be conducted to investigate the long-term effects of the Covid-19 lockdown measures. In restructuring the discussion section, we rephrased this question to become a call for future longitudinal research.

Pg 32. I am not sure where in the discussion section the authors compared their results with the results of other studies. I recommend the authors to reconsider this matter and rewrite the discussion somewhat so readers can understand the results in a broader context.

We have rewritten the discussion section, and made the comparison with the results of other studies more explicit. For example, we now explicitly compare our findings on the decrease in leisure to an initial Eurofound study, and our findings on a decrease in gender inequality in the division of household work and childcare with findings from the US.

Line 576. Future research and limitations seem to be mingled. Please rewrite.

We have disentangled these sub-sections and re-written them. 

Line 587. The authors discuss the strengths of the study here. I would suggest to mention these earlier in the discussion section before the limitations.

We have rewritten this part of the discussion and been very explicit about the key strength of our study: “A key strength of the findings presented here is that they provide evidence of the impact of an unprecedented lockdown based on a representative, probability-based sample among Dutch parents with a high response rate, thereby adding much-needed evidence to an international trend in which inequalities along gender and class lines are worsening [9,14,17–19].”

Pg 35. Line 601. “Studies such as ours demonstrate…”. Since it was an exploratory study, I would suggest the authors to formulate be a little more careful here”.

We have changed ‘demonstrate’ to ‘suggest’.

Line 611. I guess the following part of the text are concluding words. I miss a general conclusion, also in line with the purpose of the study.

We now provide a sub-section ‘General Conclusion’ at the end of the Discussion.

Final thoughts

I can imagine that the number of comments is somewhat disappointing. I hope the authors can appreciate this as a way to show that the study is an important contributor to the impact of the COVID-19 pandemic and that the comments are intended to make the manuscript easier to read. 

We very much appreciate the time taken to review our manuscript in such detail. We agree that taking these comments into consideration has significantly improved the readability of the manuscript.

---

## [Editor Report · Decision Letter 1]

30 Oct 2020

'Intelligent’ lockdown, intelligent effects? Results from a survey on gender (in)equality in paid work, the division of childcare and household work, and quality of life among parents in the Netherlands during the Covid-19 lockdown

PONE-D-20-22569R1

Dear Dr. Yerkes,

We’re pleased to inform you that your manuscript has been judged scientifically suitable for publication and will be formally accepted for publication once it meets all outstanding technical requirements.

Kind regards,

Srinivas Goli, Ph.D.

Academic Editor

PLOS ONE

Additional Editor Comments (optional):

Revisions are satisfactory to me. 
---

## [Editor Report · Acceptance letter]

12 Nov 2020

PONE-D-20-22569R1 

‘Intelligent’ lockdown, intelligent effects? Results from a survey on gender (in)equality in paid work, the division of childcare and household work, and quality of life among parents in the Netherlands during the Covid-19 lockdown 

Dear Dr. Yerkes:

I'm pleased to inform you that your manuscript has been deemed suitable for publication in PLOS ONE. Congratulations! Your manuscript is now with our production department. 

Kind regards, 

on behalf of

Dr. Srinivas Goli 

Academic Editor

PLOS ONE